# Black-box Backdoor Defense via Zero-shot Image Purification

**Yucheng Shi**[1]  **Mengnan Du**[2]  **Xuansheng Wu**[1]  **Zihan Guan**[1]  **Jin Sun**[1]  **Ninghao Liu**[1]*

[1]School of Computing, University of Georgia
[2]Department of Data Science, New Jersey Institute of Technology
{yucheng.shi, xuansheng.wu, zihan.guan, jinsun, ninghao.liu}@uga.edu
mengnan.du@njit.edu

## Abstract

Backdoor attacks inject poisoned samples into the training data, resulting in the misclassification of the poisoned input during a model's deployment. Defending against such attacks is challenging, especially for real-world black-box models where only query access is permitted. In this paper, we propose a novel defense framework against backdoor attacks through Zero-shot Image Purification (ZIP). Our framework can be applied to poisoned models without requiring internal information about the model or any prior knowledge of the clean/poisoned samples. Our defense framework involves two steps. First, we apply a linear transformation (e.g., blurring) on the poisoned image to destroy the backdoor pattern. Then, we use a pre-trained diffusion model to recover the missing semantic information removed by the transformation. In particular, we design a new reverse process by using the transformed image to guide the generation of high-fidelity purified images, which works in zero-shot settings. We evaluate our ZIP framework on multiple datasets with different types of attacks. Experimental results demonstrate the superiority of our ZIP framework compared to state-of-the-art backdoor defense baselines. We believe that our results will provide valuable insights for future defense methods for black-box models. Our code is available at `https://github.com/sycny/ZIP`.

## 1  Introduction

Machine learning has been increasingly integrated into real-world applications such as healthcare [45], finance [28] and computer vision [66]. Despite great success, machine learning models are susceptible to adversaries such as backdoor attacks [15, 8, 43, 54, 17], which compromises model security and reliability. Backdoor attacks can manipulate a model's behavior by injecting malicious samples into the training data or altering the model's weights. Although many defense strategies have been proposed to mitigate backdoor attacks, most of them require access to the model's internal structure and poisoned training data [32]. Deploying these defenses is challenging in real-world black-box scenarios, where defenders do not have access to verify or audit the inner workings of the model [18, 19]. For example, many developers and end-users now prefer using machine learning as a service (MLaaS), relying on models provided by third-party vendors for their applications. However, these models may contain backdoors, and due to copyright concerns, the services typically operate in a black-box setting with query-only access. In such scenarios, detecting and mitigating backdoor attacks is very difficult.

Currently, there are only a few backdoor defense methods that work in black-box settings. They can be categorized into two types: detecting-based and purification-based. Detecting-based methods [65,

---

*Corresponding Author

37th Conference on Neural Information Processing Systems (NeurIPS 2023).

18, 31, 12, 56] can detect the poisoned sample but could not remove the poisoned patterns. These methods are not applicable when critical poisoned samples must be used by downstream classification models. Purification-based methods can address this problem since they aim to retrieve clean images given the poisoned images. However, state-of-the-art purification approaches rely on masking and reconstructing the poisoned area [42, 53]. It can only protect against patch-based attacks. Other purification-based methods that employ image transformations for defense have the potential to defend against more sophisticated attacks but may result in a reduction in classification accuracy due to the loss of semantic information [35].

To overcome these challenges, we propose a novel framework to defend against attacks through *Zero-shot Image Purification* (ZIP). We define "*purification*" as the process of maximizing the retention of important semantic information while eliminating the trigger pattern. With this goal, we preserve the classification accuracy while breaking the connection between trigger patterns and poisoned labels. We also define "*zero-shot*" as the ability to defend against various attacks without relying on prior knowledge of attack methods. In other words, our approach does not require access to any clean or poisoned image samples (patch or non-patch based), and could be applied directly to unseen attack scenarios. This setting is crucial because real-world users usually have limited information, while new threats continue to emerge. Our proposed framework contains two main stages: we first utilize image transformation to destruct the trigger pattern, and then leverage an off-the-shelf, pre-trained diffusion generative model to restore the semantic information. Our defense strategy is based on the motivation that the semantic information in a poisoned image (e.g., faces, cars, or buildings) constitutes the majority of the data and typically falls within the training data distribution of a pre-trained image generation model. In contrast, engineered trigger patterns (e.g., mosaic patches and colorful noise) are subtle and unlikely to exist in the pre-training datasets [32, 34]. Since the diffusion model learns to sample images from the training distribution [22], purified images generated from the diffusion model will retain only their semantic information while eliminating the trigger patterns. As a result, our purification approach can effectively defend against various attacks while maintaining high-fidelity in the restored images.

Our main contributions are summarized as follows. (1) We develop a novel defense framework that can be applied to black-box models without requiring any internal information about the model. Our method is versatile and easy to use without retraining. (2) The proposed framework is designed for zero-shot settings, which do not require any prior knowledge of the clean or poisoned images. This feature relieves end-users from the need to collect samples, thus enhancing the framework's applicability. (3) Our defense framework achieves good classification accuracy on the purified images, which were originally poisoned samples, even when using an attacked model as the classifier. This improvement further enhances the framework's effectiveness and usability.

## 2 Preliminaries

### 2.1 Problem Definition

This paper addresses the backdoor defense problem in the context of image classification tasks. The goal of image classification is to learn a function $f_\theta(\mathbf{x})$ that maps input images $\mathbf{x} \in \mathcal{X}$ to their correct labels $y \in \mathcal{Y}$, where $\mathcal{X}$ denotes the image space, $\mathcal{Y}$ denotes the label space, and $\theta$ represents model parameters. Typical backdoor attacks include poisoning training data and perturbing model weights, and we use $f_\theta^{attack}$ to denote the model that has been attacked. During inference, an attacker can take a clean sample $\mathbf{x}$ and manipulate it to create a backdoor sample $\mathbf{x}^P$, e.g., adding the trigger pattern $\mathbf{p}$ to make $\mathbf{x}^P = \mathbf{x} + \mathbf{p}$. The backdoored model will misclassify $\mathbf{x}^P$ as the target label $y^{target} = f_\theta^{attack}(\mathbf{x}^P) \neq y$. Our **threat model** is in a challenging black-box setting, where defenders can only query the poisoned model and have no access to the model's internal parameters or training datasets. The attackers can modify model components or any other necessary information to implement their attacks. We formally define our backdoor defense problem as below.

**Problem 1.** *Image Purification for Backdoor Defense. Our defense is implemented in the model inference stage. Let $f_\theta^{attack}$ denote the attacked model, whose parameters are not accessible. Given a poisoned image $\mathbf{x}^P$, our goal is to obtain a purified image $\mathbf{x}^*$ from $\mathbf{x}^P$ by removing the effect of trigger $\mathbf{p}$. The purified image should be classified as the same category as the original clean image $\mathbf{x}$, i.e., $f_\theta(\mathbf{x}^*) = f_\theta(\mathbf{x}) \neq f_\theta(\mathbf{x}^P)$.*

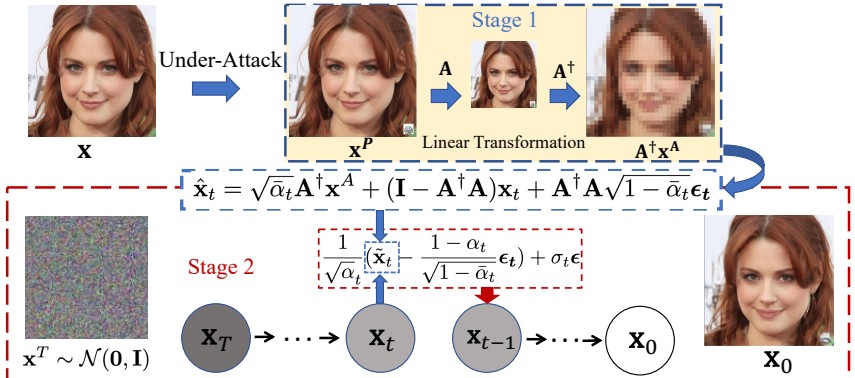

Figure 1: The proposed **ZIP** backdoor defense framework. In Stage 1, we use a linear transformation such as blurring to destruct the trigger pattern in poisoned image $\mathbf{x}^P$. In Stage 2, we design a guided diffusion process to generate the purified image $\mathbf{x}_0$ with a pre-trained diffusion model. Finally, in $\mathbf{x}_0$, the semantic information from $\mathbf{x}$ is kept while the trigger pattern is destroyed.

## 2.2 Diffusion Models

We leverage the reverse process of diffusion models to purify images. The denoising diffusion probabilistic model (DDPM) [22] is a powerful generative model for generating high-quality images. It has two processes: a forward process and a reverse process. In the forward process, the model iteratively adds noise to an input image $\mathbf{x}_0$ until it becomes random Gaussian noises $\mathbf{x}_T$; in the reverse process, the model iteratively removes the added noise from $\mathbf{x}_T$ to recover the noise-free image $\mathbf{x}_0$. More details can be found in the DDPM paper [22].

**Forward Process:** A noise-free image $\mathbf{x}_0$ is transformed to a noisy image $\mathbf{x}_t$ with controlled noise. Specifically, Gaussian noise $\boldsymbol{\epsilon}$ is gradually added to image $\mathbf{x}_0$ in $T$ steps based on a variance schedule $\beta_t \in [0, 1]$ such that $q\left(\mathbf{x}_t \mid \mathbf{x}_{t-1}\right) := \mathcal{N}\left(\mathbf{x}_t; \sqrt{1-\beta_t}\mathbf{x}_{t-1}, \beta_t\mathbf{I}\right)$, where $q$ denotes the posterior probability of $\mathbf{x}_t$ conditioned on $\mathbf{x}_{t-1}$. A nice property of this process is that the $t$-th step noisy image $\mathbf{x}_t$ could be directly generated by:

$$q\left(\mathbf{x}_t \mid \mathbf{x}_0\right) = \mathcal{N}\left(\mathbf{x}_t; \sqrt{\bar{\alpha}_t}\mathbf{x}_0, (1-\bar{\alpha}_t)\mathbf{I}\right), \quad \mathbf{x}_t = \sqrt{\bar{\alpha}_t}\mathbf{x}_0 + \sqrt{1-\bar{\alpha}_t}\boldsymbol{\epsilon}, \tag{1}$$

where $\boldsymbol{\epsilon} \sim \mathcal{N}(0, \mathbf{I})$, $\alpha_t = 1 - \beta_t$, and $\bar{\alpha}_t = \prod_{i=1}^t \alpha_i$.

**Reverse Process:** The noisy input image $\mathbf{x}_T$ is transformed into a noise-free output image $\mathbf{x}_0$ over time steps. In each step, the diffusion model takes the current image state $\mathbf{x}_t$ as input and produces the previous state $\mathbf{x}_{t-1}$. We aim to obtain clean images $\mathbf{x}_0$ by iteratively sampling $\mathbf{x}_{t-1}$ from $p(\mathbf{x}_{t-1}|\mathbf{x}_t, \mathbf{x}_0)$:

$$\mathbf{x}_{t-1} \leftarrow \frac{\sqrt{\bar{\alpha}_{t-1}}\beta_t}{1-\bar{\alpha}_t}\mathbf{x}_0 + \frac{\sqrt{\alpha_t}\left(1-\bar{\alpha}_{t-1}\right)}{1-\bar{\alpha}_t}\mathbf{x}_t + \sigma_t\boldsymbol{\epsilon}, \quad \boldsymbol{\epsilon} \sim \mathcal{N}(0, \mathbf{I}), \quad \sigma_t^2 = \frac{1-\bar{\alpha}_{t-1}}{1-\bar{\alpha}_t}\beta_t. \tag{2}$$

Based on Equation 1, the original clean image $\mathbf{x}_0$ could be approximated based on the $t$-th step observation $\mathbf{x}_t$ as: $\mathbf{x}_{0|t} = \frac{1}{\sqrt{\bar{\alpha}_t}}(\mathbf{x}_t - \sqrt{1-\bar{\alpha}_t}\boldsymbol{\epsilon}_t)$, where $\boldsymbol{\epsilon}_t$ denotes the estimation of the real $\boldsymbol{\epsilon}$ in step $t$. In each step $t$, DDPM utilizes a neural network $g_\phi(\cdot)$ to predict the noise $\boldsymbol{\epsilon}_t$, i.e., $\boldsymbol{\epsilon}_t = g_\phi(\mathbf{x}_t, t)$. With this estimation, we can convert Equation 2 into the following form as reverse process:

$$\mathbf{x}_{t-1} \leftarrow \frac{1}{\sqrt{\alpha_t}}(\mathbf{x}_t - \frac{1-\alpha_t}{\sqrt{1-\bar{\alpha}_t}}\boldsymbol{\epsilon}_t) + \sigma_t\boldsymbol{\epsilon}. \tag{3}$$

## 3 Zero-shot Image Purification (ZIP)

### 3.1 Overview of Proposed Framework

Our defense framework consists of two primary stages: (1) destruct poisoned images through image transformation, and (2) recover images using a diffusion model, as depicted in Figure 1. The first stage is based on the observation that the integrity of trigger patterns is crucial for backdoor attacks to deceive the model. Thus, destructing the trigger pattern would significantly reduce the effectiveness

of backdoor attacks [34, 46]. On the other hand, a strong transformation may destroy both the trigger pattern and semantic information. To address this, we introduce the second stage to recover the semantic information through the reverse process of diffusion models. However, traditional diffusion models (e.g., DDPM [22]) or image restoration models (e.g., DDRM [29] and DDNM [58]) can not generate high-fidelity and clean images due to a lack of direct control over the generated output.

To bridge the gap, in the following section, we first propose an image generation constraint based on the image transformation in Section 3.2. We then apply this constraint to guide the image reverse process of the diffusion model. In addition, we discuss the guide adaptation when applied in the zero-shot settings, along with its theoretical justification in Section 3.3. Finally, we introduce our efforts to improve the inference speed in Section 3.4.

## 3.2 Image Transformation and Decomposition

In the first stage, we apply image transformation to destruct potential trigger patterns, such as using average pooling to blur the poisoned images. Formally, we denote the transformation as a linear operator $\mathbf{A}$, and let the transformed image be $\mathbf{x}^A = \mathbf{A}\mathbf{x}^P = \mathbf{A}(\mathbf{x} + \mathbf{p})$. Previous research [34, 47] has used the transformed image directly as the purified result. However, such approaches may result in poor classification accuracy due to the loss of fidelity induced by $\mathbf{A}$.

To recover the lost information, an intuitive way is to apply an image generative model, e.g., the diffusion model, to yield a purified image. However, the vanilla diffusion model generates images from random Gaussian noise and lacks control over the fidelity of generated images. Thus, we propose a constraint to guide the generation process of diffusion models to recover high-fidelity images. Specifically, for an ideally purified image $\mathbf{x}_0$, it should satisfy $\mathbf{x}_0 = \mathbf{x}$, so we have:

$$\mathbf{A}(\mathbf{x}_0 + \mathbf{p}) = \mathbf{A}(\mathbf{x} + \mathbf{p}) = \mathbf{x}^A. \tag{4}$$

Based on the RND theory [48, 58], it is possible to decompose an image $\mathbf{x}$ into two parts using a linear operator $\mathbf{A}$ (e.g., average pooling) and its pseudo-inverse $\mathbf{A}^\dagger$ (e.g., upsampling) that satisfies $\mathbf{A}\mathbf{A}^\dagger\mathbf{A} = \mathbf{A}$. The decomposition is expressed as $\mathbf{x} = \mathbf{A}^\dagger\mathbf{A}\mathbf{x} + \left(\mathbf{I} - \mathbf{A}^\dagger\mathbf{A}\right)\mathbf{x}$, where the former part denotes the observable information in the range-space, and the latter part denotes the information in the null-space removed by transformation[1]. Bringing this decomposition to Equation 4, we have:

$$(\mathbf{x}_0 + \mathbf{p}) = \mathbf{A}^\dagger\mathbf{A}(\mathbf{x}_0 + \mathbf{p}) + \left(\mathbf{I} - \mathbf{A}^\dagger\mathbf{A}\right)(\mathbf{x}_0 + \mathbf{p}). \tag{5}$$

This equation derives a constraint for image purification to restore the original $\mathbf{x}$ as below:

$$\mathbf{x}_0 = \mathbf{A}^\dagger\mathbf{x}^A - \mathbf{A}^\dagger\mathbf{A}\mathbf{p} + \left(\mathbf{I} - \mathbf{A}^\dagger\mathbf{A}\right)\mathbf{x}_0, \tag{6}$$

accordingly, the ideally purified image could be decomposed into three parts. The first two parts are in the range space: the observable information stored in the transformed image $\mathbf{A}^\dagger\mathbf{x}^A$, and the intractable information embedded in the transformed trigger pattern $\mathbf{A}^\dagger\mathbf{A}\mathbf{p}$; the last part is in the null-space and is unobservable as it is removed by the transformation. To restore the lost information in the null-space, we utilize the observable information in the range-space as references.

## 3.3 Image Purification with Diffusion Model

### 3.3.1 Reverse Process Conditioned on Poisoned Images

Our proposed image purification is based on the reverse process of the diffusion model, which takes Gaussian noise $\mathbf{x}_T$ as input and generates a noise-free image $\mathbf{x}_0$. We use $\mathbf{x}_t$ to denote the image at time step $t$ in the reverse process of diffusion. To generate high-fidelity images, we propose a **rectified estimation** of $\mathbf{x}_t$ as $\mathbf{x}_t'$, so that it produces $\mathbf{x}_{0|t}$ that satisfies the decomposition constraint in Equation 6. The $\mathbf{x}_t'$ is computed using the following equation, which is derived from Equation 1 and 6. The detailed proof is in the Supplementary Material A.

$$\mathbf{x}_t' = \sqrt{\bar{\alpha}_t}\mathbf{A}^\dagger\mathbf{x}^A - \sqrt{\bar{\alpha}_t}\mathbf{A}^\dagger\mathbf{A}\mathbf{p} + (\mathbf{I} - \mathbf{A}^\dagger\mathbf{A})\mathbf{x}_t + \mathbf{A}^\dagger\mathbf{A}\sqrt{1 - \bar{\alpha}_t}\boldsymbol{\epsilon}_t, \tag{7}$$

---

[1]Usually, we have $\mathbf{A}^\dagger\mathbf{A} \neq \mathbf{I}$, which implies that this operation is lossy. When applying $\mathbf{A}^\dagger\mathbf{A}$ to an image, some information will be removed, making this operation irreversible. More details are in Supplementary Material C

where $\epsilon_t$ denotes the estimated noise, which is calculated using the pre-trained diffusion model $g_\phi$: $\epsilon_t = g_\phi(\mathbf{x}_t, t)$. Then, we modify the original reverse process in Equation 3 to accommodate this rectified estimation. The modified reverse process is expressed as:

$$\mathbf{x}_{t-1} \leftarrow \frac{1}{\sqrt{\alpha_t}}(\sqrt{\bar{\alpha}_t}\mathbf{A}^\dagger\mathbf{x}^A - \sqrt{\bar{\alpha}_t}\mathbf{A}^\dagger\mathbf{A}\mathbf{p} + (\mathbf{I} - \mathbf{A}^\dagger\mathbf{A})\mathbf{x}_t + \mathbf{A}^\dagger\mathbf{A}\sqrt{1-\bar{\alpha}_t}\epsilon_t - \frac{1-\alpha_t}{\sqrt{1-\bar{\alpha}_t}}\epsilon_t) + \sigma_t\epsilon, \quad (8)$$

where $\epsilon \sim \mathcal{N}(0, \mathbf{I})$ denotes Gaussian noise and we have $\sigma_t^2 = \frac{1-\bar{\alpha}_{t-1}}{1-\bar{\alpha}_t}\beta_t$.

### 3.3.2 Adapting Reverse Process to the Zero-shot Setting

In this subsection, we show how our reverse process design can be effectively applied in the zero-shot setting, where the trigger pattern $\mathbf{p}$ is unknown to defenders. We propose to omit the intractable term $\sqrt{\bar{\alpha}_t}\mathbf{A}^\dagger\mathbf{A}\mathbf{p}$ in Equation 7, and approximate $\mathbf{x}'_t$ with $\hat{\mathbf{x}}_t$ to obtain a new rectified estimation:

$$\hat{\mathbf{x}}_t = \sqrt{\bar{\alpha}_t}\mathbf{A}^\dagger\mathbf{x}^A + (\mathbf{I} - \mathbf{A}^\dagger\mathbf{A})\mathbf{x}_t + \mathbf{A}^\dagger\mathbf{A}\sqrt{1-\bar{\alpha}_t}\epsilon_t. \quad (9)$$

The intractable term $\sqrt{\bar{\alpha}_t}\mathbf{A}^\dagger\mathbf{A}\mathbf{p}$ can be omitted due to the following reasons:

- The value of $\sqrt{\bar{\alpha}_t}$ at the beginning of the reverse process is very small, making $\sqrt{\bar{\alpha}_t}\mathbf{A}^\dagger\mathbf{A}\mathbf{p}$ negligible compared to other terms. We provide an improved approximation in Section 3.3.3 for later stages when $\sqrt{\bar{\alpha}_t}$ increases.
- Backdoor attacks are generally stealthy or use more negligible patterns compared to the original images since the attacker wants to minimize the impact on the model's accuracy on legitimate data [43]. Thus, the destructed pattern $\mathbf{A}\mathbf{p}$ is usually negligible compared to $\mathbf{x}^A = \mathbf{A}(\mathbf{x} + \mathbf{p})$.
- The effect of $\mathbf{A}^\dagger\mathbf{A}\mathbf{p}$ is further reduced by selecting appropriate image transformations. Most backdoor attacks are characterized by severe high-frequency artifacts [65]. Therefore, transformations such as average pooling can remove the high-frequency information in $\mathbf{p}$.

Nevertheless, approximation in an iterative process such as diffusion can be risky because errors from the previous step can accumulate rapidly (e.g., exponentially) and lead to significant inaccuracies. Our method addresses this issue by ensuring the approximation error between each step is well-bounded theoretically. As a result, the final recovered image $\hat{\mathbf{x}}_0$ preserves the essential information of the original image, while the trigger pattern undergoes a transformation and is likely to be destroyed.

**Lemma 3.1.** *Suppose the estimated noise output by $g_\phi(\cdot)$ is Gaussian. Given $g_\phi(\mathbf{x}_t, t) = \epsilon_t$, we have $g_\phi((\mathbf{x}_t + \sqrt{\bar{\alpha}_t}\mathbf{A}^\dagger\mathbf{A}\mathbf{p}), t) = \epsilon_t + \epsilon'_t$, where $\epsilon_t, \epsilon'_t$ are also Gaussian.*

The error $\epsilon'_t$ in Lemma 3.1 is much smaller than the real estimated noise $\epsilon_t$. This is because the trigger pattern $\sqrt{\bar{\alpha}_t}\mathbf{A}^\dagger\mathbf{A}\mathbf{p}$, which is weighted by $\sqrt{\bar{\alpha}_t}$, is relatively subtle compared with the intermediate image $\mathbf{x}_t$. Additionally, the attacker-engineered trigger pattern $\mathbf{p}$ is unlikely to be present within the natural image distribution learned by the pre-trained diffusion model. Therefore, it is unlikely to activate the model $g_\phi(\cdot)$ and generate significant output.

**Theorem 3.2.** *Suppose that $g_\phi((\mathbf{x}_t + \sqrt{\bar{\alpha}_t}\mathbf{A}^\dagger\mathbf{A}\mathbf{p})) = \epsilon_t + \epsilon'_t$. We define the error at step $t$ between $\hat{\mathbf{x}}_t$ and $(\mathbf{x}_t + \sqrt{\bar{\alpha}_t}\mathbf{A}^\dagger\mathbf{A}\mathbf{p})$ as $\delta'_t$, i.e., $\delta'_t = \hat{\mathbf{x}}_t - (\mathbf{x}_t + \sqrt{\bar{\alpha}_t}\mathbf{A}^\dagger\mathbf{A}\mathbf{p})$. Let $\delta_t = \mathbf{A}\delta'_t$, we have the following bound on its norm: $\|\delta_t\| \leq \frac{(1-\bar{\alpha}_t)\sqrt{\alpha_{t+1}}}{\sqrt{1-\bar{\alpha}_{t+1}}}\|\mathbf{A}\|\|\epsilon'_{t+1}\|$.*

This theorem means that our proposed approximation introduces only limited approximation error at each time step after considering the error from the previous step. When paired with a common linear transform $\mathbf{A}$ like average pooling, its norm is relatively small [58]. Additionally, at the initial steps of the reverse process, the term $\|\epsilon'_{t+1}\|$ is also small according to Lemma 3.1, and we have $\frac{(1-\bar{\alpha}_t)\sqrt{\alpha_{t+1}}}{\sqrt{1-\bar{\alpha}_{t+1}}} \leq \frac{1-\bar{\alpha}_{t+1}+\alpha_t}{2}$, which is also a small value. Collectively, the error from our approximation is well bounded by a small number. According to this theorem, the final purified image exhibits the following property.

**Corollary 3.2.1.** *When $t = 0$, we have $\hat{\mathbf{x}}_0 = \mathbf{x}_0 + \mathbf{A}^\dagger\mathbf{A}\mathbf{p} + \delta'_0$, where $\mathbf{A}\delta'_0 = 0$.*

According to Corollary 3.2.1, our final purified image contains three parts: the ideally purified image $\mathbf{x}_0$, the altered trigger pattern $\mathbf{A}^\dagger\mathbf{A}\mathbf{p}$, and an approximation error $\delta'_0$. Therefore, compared with the original poisoned image $\mathbf{x}^P = \mathbf{x} + \mathbf{p}$ or the transformed image $\mathbf{x}^A = \mathbf{A}(\mathbf{x} + \mathbf{p})$, our purified image $\hat{\mathbf{x}}_0$ reduces the effect of the trigger pattern $\mathbf{p}$ and preserves the high fidelity of the image. This is critical for achieving good performance in the downstream image classification task. The proofs can be found in Supplementary Material A.

### 3.3.3 Improving ZIP with Theoretical Insights

Based on the above theoretical analysis, we propose two further improvements to our guided image purification framework to enhance its effectiveness, which include (1) adding a confidence score to the rectified estimation, and (2) introducing multiple transformations.

**Confidence Score.** The effectiveness of our framework depends on the confidence of the proposed rectified estimation. As the reverse process proceeds, the $\sqrt{\bar{\alpha}_t}$ value increases, making it difficult for $\sqrt{\bar{\alpha}_t}\mathbf{A}^{\dagger}\mathbf{A}\mathbf{p}$ to be neglected. Simply omitting this pattern would weaken the confidence in our proposed estimation.

To address this issue, we re-formulate the rectified estimation at step $t$ as: $\tilde{\mathbf{x}}_t = (1 - \bar{\alpha}_t^{\lambda})\hat{\mathbf{x}}_t + \bar{\alpha}_t^{\lambda}\mathbf{x}_t$, where $\mathbf{x}_t$ denotes the result of a pure diffusion model (e.g., DDPM) at step $t+1$, and $\lambda \geq 0$ is a hyper-parameter. Since $0 < \bar{\alpha}_t < 1$, when $\lambda = 0$, we obtain a pure reverse diffusion process without any rectification; when $\lambda = +\infty$, the reverse process reduces to ZIP. A nice property of $\tilde{\mathbf{x}}_t$ is that, $\bar{\alpha}_t$ increases as the reverse process proceeds ($t$ decreases), making the contribution of $\mathbf{x}_t$ larger. This is reasonable because the intermediate image becomes increasingly informative over time, reducing the reliance on $\hat{\mathbf{x}}_t$ in reverse diffusion. We set $\lambda$ value to achieve desirable fidelity scores (e.g., PSNR [29, 58]) for the resultant purified images. Finally, our revised reverse process is defined as:

$$\mathbf{x}_{t-1} \leftarrow \frac{1}{\sqrt{\alpha_t}}(\tilde{\mathbf{x}}_t - \frac{1 - \alpha_t}{\sqrt{1 - \bar{\alpha}_t}}\boldsymbol{\epsilon}_t) + \sigma_t\boldsymbol{\epsilon}. \tag{10}$$

**Multiple Transformations.** To improve the effectiveness of our approach in removing the poisoning effect, we propose including multiple transformations to more effectively destroy the unknown trigger pattern in a zero-shot setting. Specifically, given $N$ transformations, the intermediate image $\mathbf{x}_t$ and estimated noise $\boldsymbol{\epsilon}_t$ should all satisfy:

$$\hat{\mathbf{x}}_t^n = \sqrt{\bar{\alpha}_t}\mathbf{A}_n^{\dagger}\mathbf{x}^{A_n} + (\mathbf{I} - \mathbf{A}_n^{\dagger}\mathbf{A}_n)\mathbf{x}_t + \mathbf{A}_n^{\dagger}\mathbf{A}_n\sqrt{1 - \bar{\alpha}_t}\boldsymbol{\epsilon}_t, \quad n = 1, 2, ..., N. \tag{11}$$

Finally, we have $\hat{\mathbf{x}}_t = \frac{1}{N}(\hat{\mathbf{x}}_t^1 + \hat{\mathbf{x}}_t^2 + ... + \hat{\mathbf{x}}_t^N)$. We provide our proposed algorithm in Algorithm 1. We focus on two types of transformation pairs in this paper: blurring, represented by $\mathbf{A}_1^{\dagger}\mathbf{A}_1$, and gray-scale conversion, represented by $\mathbf{A}_2^{\dagger}\mathbf{A}_2$.

### 3.4 Purification Speed-up

The purification speed is crucial when performing defense at inference time. Our framework leverages pre-trained diffusion models, and conducts purification on each test image. Hence, we propose several techniques to speed up purification and reduce costs.

#### 3.4.1 Diffusion Model Inference Speed-up

Algorithm 1 requires a large number of steps to generate a single sample. Each step involves computing the estimated noise and diffusion process, which can be computationally expensive. To address this issue, we leverage the power of the denoising diffusion implicit model (DDIM) [50] to improve the inference speed of our reverse process as below:

$$\mathbf{x}_{t-1} \leftarrow \sqrt{\bar{\alpha}_{t-1}}\tilde{\mathbf{x}}_{0|t} + \sqrt{1 - \bar{\alpha}_{t-1} - \sigma_t^2}\boldsymbol{\epsilon}_t + \sigma_t\boldsymbol{\epsilon}, \tag{12}$$

where $\tilde{\mathbf{x}}_{0|t}$ is also an estimated image based on $\tilde{\mathbf{x}}_t$, and we have $\tilde{\mathbf{x}}_{0|t} = \frac{1}{\sqrt{\bar{\alpha}_t}}(\tilde{\mathbf{x}}_t - \sqrt{1 - \bar{\alpha}_t}\boldsymbol{\epsilon}_t)$. By using

---

**Algorithm 1** Zero-shot Image Purification

**Require:** Test image for purification $\mathbf{x}^P$; linear transformation $\mathbf{A}_1, \mathbf{A}_2, ..., \mathbf{A}_n$ and their pseudo-inverse $\mathbf{A}_1^{\dagger}, \mathbf{A}_2^{\dagger}, ..., \mathbf{A}_n^{\dagger}$; diffusion model $g$; hyperparameter $\lambda$.

**Ensure:** $\mathbf{x}^{A_n} = \mathbf{A}_n\mathbf{x}^P$, $n = 0, 1, ..., N$
1: $\mathbf{x}^T \sim \mathcal{N}(\mathbf{0}, \mathbf{I})$
2: **for** $t = T, T-1, ..., 1$ **do**
3: $\quad \boldsymbol{\epsilon} \sim \mathcal{N}(\mathbf{0}, \mathbf{I})$ if $t > 1$, else $\boldsymbol{\epsilon} = \mathbf{0}$
4: $\quad \boldsymbol{\epsilon}_t = g_{\phi}(\mathbf{x}_t, t)$
5: $\quad$ **for** $n = 1, 2, ..., N$ **do**
6: $\quad\quad \hat{\mathbf{x}}_t^n = \sqrt{\bar{\alpha}_t}\mathbf{A}_1^{\dagger}\mathbf{x}^{A_n} + (\mathbf{I} - \mathbf{A}_n^{\dagger}\mathbf{A}_n)\mathbf{x}_t + \mathbf{A}_n^{\dagger}\mathbf{A}_n\sqrt{1 - \bar{\alpha}_t}\boldsymbol{\epsilon}_t$
7: $\quad$ **end for**
8: $\quad \hat{\mathbf{x}}_t = \frac{1}{N}(\hat{\mathbf{x}}_t^1 + \hat{\mathbf{x}}_t^2 +, ..., +\hat{\mathbf{x}}_t^N)$
9: $\quad \tilde{\mathbf{x}}_t = (1 - \bar{\alpha}_t^{\lambda})\hat{\mathbf{x}}_t + \bar{\alpha}_t^{\lambda}\mathbf{x}_t$
10: $\quad \mathbf{x}_{t-1} \leftarrow \frac{1}{\sqrt{\alpha_t}}(\tilde{\mathbf{x}}_t - \frac{1-\alpha_t}{\sqrt{1-\bar{\alpha}_t}}\boldsymbol{\epsilon}_t) + \sigma_t\boldsymbol{\epsilon}$
11: **end for**
12: **return** $\mathbf{x}_0$

---

this speed-up inference method, we can obtain a high-fidelity purified image by sampling only a few steps instead of conducting sampling in thousands of steps. The modified algorithm based on DDIM is provided in the Supplementary Material D.

### 3.4.2 Framework Speed-up

**Batch Data Speed-up.** We introduce an acceleration method for purification, when images can be processed in batch during inference. For instance, if the pre-trained model accepts images of size $L \times L$ as input, and we wish to purify $l \times l$ images, we can first combine $(L/l)^2$ images into a single $L \times L$ image by tiling. We then apply our purification method to the tiled image, and split the purified image back to the original images afterwards. Larger images could be resized into $l \times l$ to apply this method. In this way, our defense approach can handle images of various sizes and significantly improve the purification speed.

**Streaming Data Speed-up.** In the streaming data scenarios, defenders may only have access to a single test image at a time. In such cases, we can use a fast zero-shot detection method such as [19] to quickly assess whether the image is likely to be poisoned. If the detection result indicates a potential poisoning, we can apply our purification method to remove the trigger pattern. This approach allows us to quickly and effectively defend against poisoned images without applying our purification on every image, thus saving the average processing time.

## 4 Experiments

### 4.1 Experimental Settings

For defense evaluation, we conducted experiments on three types of backdoor attacks: BadNet [15], Attack in the physical world (PhysicalBA) [34], and Blended [8]. The first two attacks are representatives of patch-based attacks, while the last one represents a non-patch-based backdoor attack. To configure these attack algorithms, we follow the benchmark setting in [33] and use the provided codes. We evaluate the effectiveness of our defense framework ZIP on three datasets: CIFAR-10 [30], GTSRB [52], and Imagenette [24]. The poisoned classification network is based on ResNet-34 [21]. Specifically, for the CIFAR-10 dataset, we apply both blur and gray-scale conversion as linear transformations. For the GTSRB and Imagenette datasets, we solely apply blur as the linear transformation. The additional experiment details are in the Supplementary Material E.

Defense methods available for black-box purification in zero-shot are rare. To fairly assess the effectiveness of our proposed method, we conduct a comparative evaluation against two baseline approaches: ShrinkPad [35] and image blurring (referred to as Blur). The former is a state-of-the-art image transformation-based defense method that can work on black-box models in the zero-shot setting. The latter uses blurred images $\mathbf{A}_1^\dagger \mathbf{x}^{A_1}$ as purified images. We apply defense methods to all test samples, and then evaluate the output using a poisoned classification network. In addition to the clean accuracy (CA) and attack success rate (ASR) metrics for assessing defense effectiveness, we introduce a new metric called poisoned accuracy (PA). It measures the classification performance of the purified poisoned samples. A higher value of PA indicates that the purified poisoned samples are more likely to be correctly classified, even when using an attacked classification model.

### 4.2 Qualitative Results of Purification

We conduct qualitative case studies (see Figure 2) of our method in purifying poisoned images created by the Blended and BadNet attacks. We show examples of poisoned images $\mathbf{x}^P$ and purified image $\mathbf{x}_0$, where the trigger pattern is clearly visible in the poisoned images but has been altered/removed in the purified images. For comparison, we also show the blurred image $\mathbf{A}_1^\dagger \mathbf{x}^{A_1}$ and the grayscale images $\mathbf{A}_2^\dagger \mathbf{x}^{A_2}$. The transformed images can destruct the trigger pattern, but they also alter a lot of semantic information. These results demonstrate the effectiveness of ZIP in removing the effect of trigger patterns from images while maintaining semantic information. More qualitative results for different attacks are in the Supplementary Material B.

### 4.3 Quantitative Results of Defense

We conduct quantitative experiments where the results are in Table 1. Our observations are as follows: **(1)** Our method effectively defends against different backdoor attacks. This is because the transformations in our framework, which are independent of specific attacks, break the connection between backdoor triggers and backdoor labels. **(2)** Our method achieves overall better classification

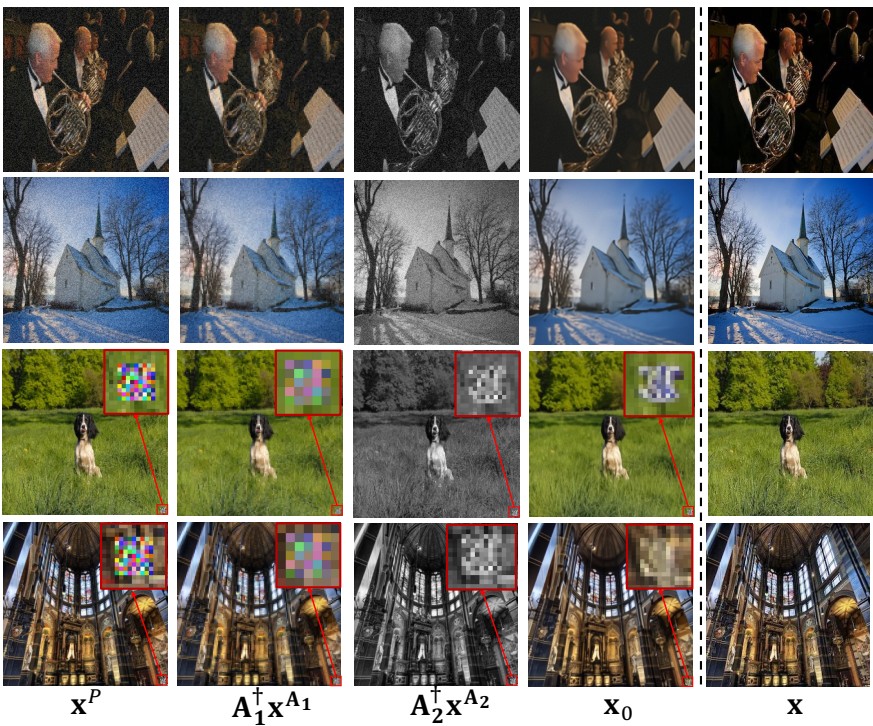

| $\mathbf{x}^P$ | $\mathbf{A}_1^\dagger \mathbf{x}^{\mathbf{A}_1}$ | $\mathbf{A}_2^\dagger \mathbf{x}^{\mathbf{A}_2}$ | $\mathbf{x}_0$ | $\mathbf{x}$ |

Figure 2: Qualitative analysis of purification. (Row 1-2: Blended; Row 3-4: BadNets. More qualitative results are listed in Supplementary Material B)

Table 1: The clean accuracy (CA %), the attack success rate (ASR %), and the poisoned accuracy (PA %) of defense methods against different backdoor attacks. *None* means no attacks are applied.

| Dataset | Attack | No Defense | | | ShrinkPad (defense) | | | Blur (defense) | | | **ZIP** (Ours) | | |
|---|---|---|---|---|---|---|---|---|---|---|---|---|---|
| | | CA ↑ | ASR ↓ | PA ↑ | CA ↑ | ASR ↓ | PA ↑ | CA ↑ | ASR ↓ | PA ↑ | CA ↑ | ASR ↓ | PA ↑ |
| CIFAR-10 (32 × 32) (10 classes) | None | 80.15 | — | — | — | — | — | — | — | — | — | — | — |
| | BadNet | 82.31 | 99.98 | 10.00 | 62.89 | 9.34 | 63.12 | 58.19 | 21.78 | 53.47 | **78.97** | **5.53** | **79.10** |
| | Blended | 80.26 | 99.96 | 10.03 | 58.97 | **2.28** | 40.22 | 55.91 | 3.04 | 49.91 | **72.62** | 7.75 | **57.98** |
| | PhysicalBA | 85.30 | 98.73 | 11.20 | **82.84** | 90.50 | 18.37 | 41.84 | **1.09** | 41.37 | 80.10 | 4.33 | **80.33** |
| | Average | 82.62 | 99.56 | 10.41 | 68.23 | 34.04 | 40.57 | 51.98 | 8.64 | 48.25 | **77.23** | 5.87 | **72.47** |
| GTSRB (32 × 32) (43 classes) | None | 96.95 | — | — | — | — | — | — | — | — | — | — | — |
| | BadNet | 96.53 | 99.99 | 5.70 | 78.33 | **5.81** | 78.82 | 95.98 | 7.33 | 95.11 | **96.18** | 6.19 | **96.03** |
| | Blended | 96.58 | 99.89 | 5.79 | 76.76 | 10.54 | 56.41 | 93.68 | 11.07 | 73.91 | **95.74** | 8.53 | **81.27** |
| | PhysicalBA | 96.83 | 100.00 | 5.70 | **97.41** | 100.00 | 5.70 | 91.00 | **5.53** | 90.53 | 95.44 | 6.57 | **94.91** |
| | Average | 96.65 | 99.96 | 5.73 | 84.17 | 38.78 | 46.98 | 93.55 | 7.98 | 86.52 | **95.79** | 7.10 | **90.74** |
| Imagenette (256 × 256) (10 classes) | None | 84.58 | — | — | — | — | — | — | — | — | — | — | — |
| | BadNet | 84.99 | 94.53 | 14.98 | 71.23 | 8.56 | 70.72 | 81.47 | 16.45 | 79.94 | **84.05** | **7.55** | **83.97** |
| | Blended | 86.14 | 99.85 | 10.19 | 74.06 | 20.63 | 36.10 | 78.95 | 79.41 | 25.57 | **81.42** | 8.35 | **78.36** |
| | PhysicalBA | 90.67 | 72.94 | 34.29 | **90.21** | 96.81 | 13.07 | 84.84 | 32.40 | 74.87 | 87.26 | 10.91 | **86.54** |
| | Average | 87.27 | 89.11 | 19.82 | 78.50 | 42.00 | 39.96 | 81.75 | 42.75 | 60.13 | **84.24** | 8.94 | **82.96** |

accuracy (CA) compared to baseline methods because our formulation successfully recovers semantic information to ensure accurate downstream classification. For example, on the Imagenette dataset, ZIP reduces the ASR of the BadNet attack from 94.53% (no defense) to just 7.55%, with only a 0.94% drop in CA. Similarly, on the GTSRB dataset, our approach reduces the success rate of the PhysicalBA attack from 100% (no defense) to 6.57%, with only a 1.39% drop in CA. **(3)** Our method outperforms the baselines in poisoned accuracy (PA), indicating that poisoned samples can still be used for classification even when using an attacked black-box classifier. It demonstrates the robustness and usability of ZIP in real-world scenarios. **(4)** ShrinkPad performs poorly on the PhysicalBA attack, which is consistent with the findings in [34]. This is because PhysicalBA is specifically designed to evade defense methods such as ShrinkPad [34]. On the other hand, our method successfully defends against this attack, demonstrating its superior performance. We include additional experiment results of defenses in Supplementary Material H.

Table 2: ZIP with different transformations (CA↑ / ASR↓ / PA↑).

| Attack | $Blur_2$ | $Blur_4$ | $Blur_8$ | Grayscale |
|---|---|---|---|---|
| BadNets | 84.05/7.55/83.97 | 82.77/7.15/82.39 | 72.38/13.41/72.50 | 70.52/10.79/68.61 |
| Blended | 84.05/26.11/63.46 | 81.42/8.35/78.36 | 71.03/7.69/67.13 | 79.79/99.40/10.59 |
| PhysicalBA | 90.24/22.88/79.89 | 87.26/10.91/86.54 | 76.58/19.43/76.43 | 76.66/17.51/74.87 |

## 4.4 Ablation Studies

### 4.4.1 Evaluation on Transformations

**Comparison of Different Transformations.** Image transformation plays a crucial role in our defense. We evaluate the effectiveness of ZIP on the Imagenette dataset using four distinct types of transformations: $Blur_2$, $Blur_4$, $Blur_8$, and Grayscale (the results are in Table 2). The subscript of "Blur" represents the kernel size of average pooling, where a larger number indicates a stronger transformation. We have the following observations: (1) The blur operation is more effective than grayscale conversion in reducing the ASR while maintaining good CA and PA. (2) Generally, stronger transformations are more effective in destructing the trigger pattern, resulting in a lower ASR. However, they also destroy more semantic information, leading to lower classification accuracy.

**Comparison to Masking and Reconstruction.** Linear transformations are more effective than masking in removing poisoned effects, particularly in cases like the Blended attack where the backdoor pattern is distributed across the image. We compare ZIP with a recently proposed purification method BDMAE [53] on CIFAR-10.

Table 3: Comparison to BDMAE (CA↑ / ASR↓).

| Attack | No Defense | BDMAE | ZIP |
|---|---|---|---|
| BadNet | 82.31/99.98 | 81.44/1.12 | 78.97/5.53 |
| Blended | 80.26/99.96 | 78.57/99.88 | 72.62/7.75 |

BDMAE first identifies the trigger region on a test image, masks the region, and then uses a masked autoencoder to restore the image. To ensure a fair comparison, we apply the defense stage of BDMAE to our benchmarks using its official code. Table 3 shows that BDMAE effectively defends against the BadNet attack, but it fails to defend against the Blended attack. This is because the trigger pattern in Blended attacks is not located in a local region, making it difficult for BDMAE to identify an appropriate mask. In contrast, our ZIP framework does not make assumptions about trigger patterns thus successfully defends against both attacks.

### 4.4.2 Evaluation on Enhanced Attacks

We consider a more challenging scenario, where we assume the attacker is aware of our defense and has access to the purified backdoor images, which are then used as enhanced poisoned images to attack a classifier. To demonstrate the effectiveness of our proposed method in defending against such enhanced attacks, we apply our method to BadNet and Blended as examples and choose blurring and grayscale conversion together as the transformations of ZIP. Other settings remain the same as the original attack, and more details about the settings are in the Supplementary Material F.1.

Table 4: Defense against enhanced attacks (CA↑ / ASR↓ / PA↑).

| Attack | Original Trigger | ShrinkPad | Blur | ZIP (Blur+Grayscale) | ZIP (Blur) | ZIP (Grayscale) |
|---|---|---|---|---|---|---|
| BadNets | 82.36/24.00/76.05 | 69.32/10.34/70.44 | 80.58/31.41/70.70 | 58.01/79.89/28.02 | 82.39/18.47/77.29 | 63.26/73.52/32.91 |
| Blended | 85.42/21.83/48.30 | 75.10/11.71/47.13 | 78.47/63.61/37.04 | 75.79/75.21/33.01 | 81.63/28.83/51.32 | 73.80/40.71/36.22 |

During the inference stage, we continue to use the original trigger pattern for attacks. From the experimental results in Table 4, we observe that the original trigger pattern no longer triggers the attack, but if the purification (Blur+Grayscale) we use has the same transformation as the attacker, the attacks can still be triggered. However, by switching to a different transformation, such as solely Blur or Grayscale, the attacks can be effectively mitigated. It is important to note that in practical scenarios, the attacker may not know which transformation the defender is using, therefore the defender should consider using diverse transformations to enhance the defense ability.

### 4.4.3 Evaluation on Purification Speed

We analyze the computational costs of classification and purification before and after applying speed-up strategies. Figure 3 presents the average time required per image in CIFAR-10. Our defense method's efficiency can be enhanced with the introduction of our proposed tiling and detection-based strategies, resulting in a $33\times$ and $39\times$ speedup, respectively. The results also show that ZIP can complete the purification step faster than the classification step under various scenarios, highlighting its efficiency. In addition, our method is three times faster than another inference time purification model BDMAE [53]. More details about the speed-up settings are provided in the Supplementary Material F.2.

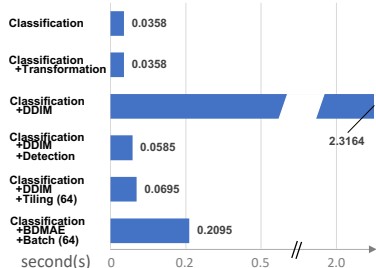

Figure 3: Inference time with different purification on CIFAR-10.

## 5 Related Work: Backdoor Defense

In this section, we briefly review backdoor defenses here and provide a more detailed discussion in the Appendix G. Existing backdoor defense methods are mainly designed for white-box models [32, 61]. However, these methods [47, 10] often require access to model parameters or original training data, which is not always feasible in real-world scenarios. To address this challenge, existing black-box defense methods are proposed and can be roughly divided into backdoor detection and backdoor purification. Detection models [18, 19, 65, 12, 38, 14, 56, 16] aim to identify and reject any detected poisoned images for further inference as a defense mechanism. However, this approach can limit the usefulness of these methods in practical settings where users expect results for all of their test samples. On the other hand, backdoor purification methods aim to remove the poison effect from the image to defend against attacks. Some methods [42, 53] in this category involve masking the potentially poisoned region and then reconstructing the masked image to obtain a poison-free image. However, these strategies may fail when the trigger patterns are distributed throughout the image, rather than in a specific patch-based location [1, 43, 68, 37]. Another approach [35] involves applying strong image transformations to the test image to destruct the trigger pattern. While such methods can defend against more advanced attacks, they typically result in a decrease in classification accuracy.

## 6 Conclusion

We propose a novel framework called ZIP for defending against backdoor attacks in black-box settings. Our method involves applying strong transformations to the poisoned image to destroy the trigger pattern. It then leverages a pre-trained diffusion model to recover the removed semantic information while maintaining the fidelity of the purified images. The experiments demonstrate the effectiveness of ZIP in defending against various backdoor attacks, without requiring model internal information or any training samples. ZIP also enables end-users to utilize full test samples, even when using an attacked classification model. Some future directions include designing black-box defense for other data domains and exploring other types of diffusion models.

## Acknowledgments and Disclosure of Funding

The work is, in part, supported by NSF (#2223768, #2310261). The views and conclusions in this paper are those of the authors and should not be interpreted as representing any funding agencies.

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

# A  Theoretical Justification

## A.1  Proof on rectified estimation of $\mathbf{x}_t$ in Equation 7

**Equation 7**  $\mathbf{x}'_t = \sqrt{\bar{\alpha}_t}\mathbf{A}^\dagger \mathbf{x}^A - \sqrt{\bar{\alpha}_t}\mathbf{A}^\dagger \mathbf{A}\mathbf{p} + (\mathbf{I} - \mathbf{A}^\dagger \mathbf{A})\mathbf{x}_t + \mathbf{A}^\dagger \mathbf{A}\sqrt{1 - \bar{\alpha}_t}\boldsymbol{\epsilon}_t.$

*Proof.*  Based on $\mathbf{x}_t = \sqrt{\bar{\alpha}_t}\mathbf{x}_0 + \sqrt{1 - \bar{\alpha}_t}\boldsymbol{\epsilon}$, we have a nice property [22]:

$$\mathbf{x}_{0|t} = \frac{1}{\sqrt{\bar{\alpha}_t}}(\mathbf{x}_t - \sqrt{1 - \bar{\alpha}_t}\boldsymbol{\epsilon}_t). \tag{13}$$

To ensure that the approximated clean image $\mathbf{x}_{0|t}$ based on the $t$-th step observation $\mathbf{x}_t$ satisfies the constraint in Equation 6, we have:

$$\mathbf{x}_{0|t} = \mathbf{A}^\dagger \mathbf{x}^A - \mathbf{A}^\dagger \mathbf{A}\mathbf{p} + \left(\mathbf{I} - \mathbf{A}^\dagger \mathbf{A}\right)\mathbf{x}_{0|t}. \tag{14}$$

Combine the above two equations, we can have:

$$\frac{1}{\sqrt{\bar{\alpha}_t}}(\mathbf{x}_t - \sqrt{1 - \bar{\alpha}_t}\boldsymbol{\epsilon}_t) = \mathbf{A}^\dagger \mathbf{x}^A - \mathbf{A}^\dagger \mathbf{A}\mathbf{p} + \left(\mathbf{I} - \mathbf{A}^\dagger \mathbf{A}\right)\left(\frac{1}{\sqrt{\bar{\alpha}_t}}(\mathbf{x}_t - \sqrt{1 - \bar{\alpha}_t}\boldsymbol{\epsilon}_t)\right). \tag{15}$$

Taking the derivative, we arrive at:

$$\mathbf{x}'_t = \sqrt{\bar{\alpha}_t}\mathbf{A}^\dagger \mathbf{x}^A - \sqrt{\bar{\alpha}_t}\mathbf{A}^\dagger \mathbf{A}\mathbf{p} + (\mathbf{I} - \mathbf{A}^\dagger \mathbf{A})\mathbf{x}_t + \mathbf{A}^\dagger \mathbf{A}\sqrt{1 - \bar{\alpha}_t}\boldsymbol{\epsilon}_t. \tag{16}$$

$\square$

## A.2  Proof of Lemma 3.1

**Lemma 3.1** *Suppose the estimated noise output by $g_\phi(\cdot)$ is Gaussian. Given $g_\phi(\mathbf{x}_t, t) = \boldsymbol{\epsilon}_t$, we have $g_\phi((\mathbf{x}_t + \sqrt{\bar{\alpha}_t}\mathbf{A}^\dagger \mathbf{A}\mathbf{p}), t) = \boldsymbol{\epsilon}_t + \boldsymbol{\epsilon}'_t$, where $\boldsymbol{\epsilon}_t, \boldsymbol{\epsilon}'_t$ are also Gaussian.*

*Proof.*  Let us define the output of $g_\phi((\mathbf{x}_t + \mathbf{x}'_t), t)$ as $\hat{\boldsymbol{\epsilon}}_t$, so we have $g_\phi((\mathbf{x}_t + \mathbf{x}'_t), t) = \hat{\boldsymbol{\epsilon}}_t$. Next, we define $\boldsymbol{\epsilon}_t \sim N(\mu_1, \sigma_1^2)$ and $\hat{\boldsymbol{\epsilon}}_t \sim N(\mu_2, \sigma_2^2)$.

Since $\hat{\boldsymbol{\epsilon}}_t$ also follows a Gaussian distribution, we can subtract $\boldsymbol{\epsilon}_t$ from $\hat{\boldsymbol{\epsilon}}_t$ to obtain $\boldsymbol{\epsilon}'_t$, such that:

$$\boldsymbol{\epsilon}'_t = \hat{\boldsymbol{\epsilon}}_t - \boldsymbol{\epsilon}_t \sim N(\mu_2 - \mu_1, \sigma_1^2 + \sigma_2^2) \tag{17}$$

This confirms that $\boldsymbol{\epsilon}'_t$ is also Gaussian, thus completing the proof.  $\square$

## A.3  Proof of Theorem 3.2

**Theorem 3.2** *Suppose that $g_\phi((\mathbf{x}_t + \sqrt{\bar{\alpha}_t}\mathbf{A}^\dagger \mathbf{A}\mathbf{p})) = \boldsymbol{\epsilon}_t + \boldsymbol{\epsilon}'_t$. We define the error at step $t$ between $\hat{\mathbf{x}}_t$ and $(\mathbf{x}_t + \sqrt{\bar{\alpha}_t}\mathbf{A}^\dagger \mathbf{A}\mathbf{p})$ as $\boldsymbol{\delta}'_t$, i.e., $\boldsymbol{\delta}'_t = \hat{\mathbf{x}}_t - (\mathbf{x}_t + \sqrt{\bar{\alpha}_t}\mathbf{A}^\dagger \mathbf{A}\mathbf{p})$. Let $\boldsymbol{\delta}_t = \mathbf{A}\boldsymbol{\delta}'_t$, we have the following bound on its norm: $\|\boldsymbol{\delta}_t\| \leq \frac{(1-\bar{\alpha}_t)\sqrt{\alpha_{t+1}}}{\sqrt{1-\bar{\alpha}_{t+1}}}\|\mathbf{A}\|\|\boldsymbol{\epsilon}'_{t+1}\|$.*

*Proof.*  We prove the above theorem by induction.

1  The base case is when $t = T$, where we have $\hat{\mathbf{x}}_T - (\mathbf{x}_T + \sqrt{\bar{\alpha}_T}\mathbf{A}^\dagger \mathbf{A}\mathbf{p}) = 0$, which holds.

2  Suppose for $\hat{\mathbf{x}}_t - (\mathbf{x}_t + \sqrt{\bar{\alpha}_t}\mathbf{A}^\dagger \mathbf{A}\mathbf{p}) = \mathbf{A}^\dagger \boldsymbol{\delta_t}, \|\boldsymbol{\delta_t}\| \leq \frac{(1-\bar{\alpha}_t)\sqrt{\alpha_{t+1}}}{\sqrt{1-\bar{\alpha}_{t+1}}}\|\mathbf{A}\|\|\boldsymbol{\epsilon}'_{t+1}\|$ is true.

3  Induction:

$$\mathbf{x}_{t-1} \leftarrow \frac{1}{\sqrt{\alpha_t}}(\sqrt{\bar{\alpha}_t}\mathbf{A}^\dagger \mathbf{x}^A - \sqrt{\bar{\alpha}_t}\mathbf{A}^\dagger \mathbf{A}\mathbf{p} + (\mathbf{I} - \mathbf{A}^\dagger \mathbf{A})\mathbf{x}_t + \mathbf{A}^\dagger \mathbf{A}\sqrt{1 - \bar{\alpha}_t}\boldsymbol{\epsilon}_t - \frac{1 - \alpha_t}{\sqrt{1 - \bar{\alpha}_t}}\boldsymbol{\epsilon}_t) + \sigma_t\boldsymbol{\epsilon}, \tag{18}$$

$$\hat{\mathbf{x}}_{t-1} = \frac{1}{\sqrt{\alpha_t}}(\sqrt{\bar{\alpha}_t}\mathbf{A}^\dagger \mathbf{x}^A + (\mathbf{I} - \mathbf{A}^\dagger \mathbf{A})\hat{\mathbf{x}}_t + \mathbf{A}^\dagger \mathbf{A}\sqrt{1 - \bar{\alpha}_t}(\boldsymbol{\epsilon}_t + \boldsymbol{\epsilon}'_t) - \frac{1 - \alpha_t}{\sqrt{1 - \bar{\alpha}_t}}(\boldsymbol{\epsilon}_t + \boldsymbol{\epsilon}'_t)) + \sigma_t\boldsymbol{\epsilon}, \tag{19}$$

$$\hat{\mathbf{x}}_{t-1} - \mathbf{x}_{t-1} \leftarrow \frac{1}{\sqrt{\alpha_t}}(\sqrt{\bar{\alpha}_t}\mathbf{A}^\dagger\mathbf{A}\mathbf{p} + (\mathbf{I} - \mathbf{A}^\dagger\mathbf{A})(\sqrt{\bar{\alpha}_t}\mathbf{A}^\dagger\mathbf{A}\mathbf{p} + \mathbf{A}^\dagger\boldsymbol{\delta_t}) + \mathbf{A}^\dagger\mathbf{A}\sqrt{1 - \bar{\alpha}_t}\boldsymbol{\epsilon_t'} - \frac{1 - \alpha_t}{\sqrt{1 - \bar{\alpha}_t}}\boldsymbol{\epsilon_t'})$$
(20)

By definition, we have $\hat{\mathbf{x}}_{t-1} - (\mathbf{x}_{t-1} + \sqrt{\bar{\alpha}_{t-1}}\mathbf{A}^\dagger\mathbf{A}\mathbf{p}) = \boldsymbol{\delta'_{t-1}}$. Let $\boldsymbol{\delta_{t-1}} = \mathbf{A}\boldsymbol{\delta'_{t-1}}$, we have $\boldsymbol{\delta_{t-1}} = \mathbf{A}\boldsymbol{\delta'_{t-1}} = \mathbf{A}(\hat{\mathbf{x}}_{t-1} - (\mathbf{x}_{t-1} + \sqrt{\bar{\alpha}_{t-1}}\mathbf{A}^\dagger\mathbf{A}\mathbf{p}))$. Since $\mathbf{A}(\mathbf{I} - \mathbf{A}^\dagger\mathbf{A}) = \mathbf{0}$, we can have:

$$\boldsymbol{\delta_{t-1}} = \frac{1}{\sqrt{\alpha_t}}(\mathbf{A}\sqrt{1 - \bar{\alpha}_t}\boldsymbol{\epsilon_t'} - \mathbf{A}\frac{1 - \alpha_t}{\sqrt{1 - \bar{\alpha}_t}}\boldsymbol{\epsilon_t'}) = \frac{(1 - \bar{\alpha}_{t-1})\sqrt{\alpha_t}}{\sqrt{1 - \bar{\alpha}_t}}\mathbf{A}\boldsymbol{\epsilon_t'}$$
(21)

Based on the Cauchy–Schwarz inequality:

$$\|\boldsymbol{\delta_{t-1}}\| = \|\frac{(1 - \bar{\alpha}_{t-1})\sqrt{\alpha_t}}{\sqrt{1 - \bar{\alpha}_t}}\mathbf{A}\boldsymbol{\epsilon_t'}\| \leq \frac{(1 - \bar{\alpha}_{t-1})\sqrt{\alpha_t}}{\sqrt{1 - \bar{\alpha}_t}}\|\mathbf{A}\|\|\boldsymbol{\epsilon_t'}\|$$
(22)

$\square$

## A.4   Proof of Corollary 3.2.1

**Corollary 3.2.1** *When $t = 0$, we have $\hat{\mathbf{x}}_0 = \mathbf{x}_0 + \mathbf{A}^\dagger\mathbf{A}\mathbf{p} + \boldsymbol{\delta'_0}$, where $\mathbf{A}\boldsymbol{\delta'_0} = \mathbf{0}$.*

*Proof.*  First, we have:

$$\mathbf{x}_0 \leftarrow \frac{1}{\sqrt{\alpha_1}}(\sqrt{\bar{\alpha}_1}\mathbf{A}^\dagger\mathbf{x}^A - \sqrt{\bar{\alpha}_1}\mathbf{A}^\dagger\mathbf{A}\mathbf{p} + (\mathbf{I} - \mathbf{A}^\dagger\mathbf{A})\mathbf{x}_1 + \mathbf{A}^\dagger\mathbf{A}\sqrt{1 - \bar{\alpha}_1}\boldsymbol{\epsilon}_1 - \frac{1 - \alpha_1}{\sqrt{1 - \bar{\alpha}_1}}\boldsymbol{\epsilon}_1) + \sigma_1\boldsymbol{\epsilon}, \text{ (23)}$$

$$\hat{\mathbf{x}}_0 = \frac{1}{\sqrt{\alpha_1}}(\sqrt{\bar{\alpha}_1}\mathbf{A}^\dagger\mathbf{x}^A + (\mathbf{I} - \mathbf{A}^\dagger\mathbf{A})\hat{\mathbf{x}}_1 + \mathbf{A}^\dagger\mathbf{A}\sqrt{1 - \bar{\alpha}_1}(\boldsymbol{\epsilon}_1 + \boldsymbol{\epsilon_1'}) - \frac{1 - \alpha_1}{\sqrt{1 - \bar{\alpha}_1}}(\boldsymbol{\epsilon}_1 + \boldsymbol{\epsilon_1'})) + \sigma_1\boldsymbol{\epsilon}, \text{ (24)}$$

Then we can have:

$$\mathbf{A}\boldsymbol{\delta'_0} = \frac{(1 - \alpha_0)\sqrt{\alpha_1}}{\sqrt{1 - \bar{\alpha}_1}}\mathbf{A}\boldsymbol{\epsilon_1'}.$$
(25)

Since $\alpha_0 = 1$, then we have $\mathbf{A}\boldsymbol{\delta'_0} = \mathbf{0}$.  $\square$

# B   Qualitative Results of Purification

## B.1   Qualitative Results of Purification on BadNet Attack

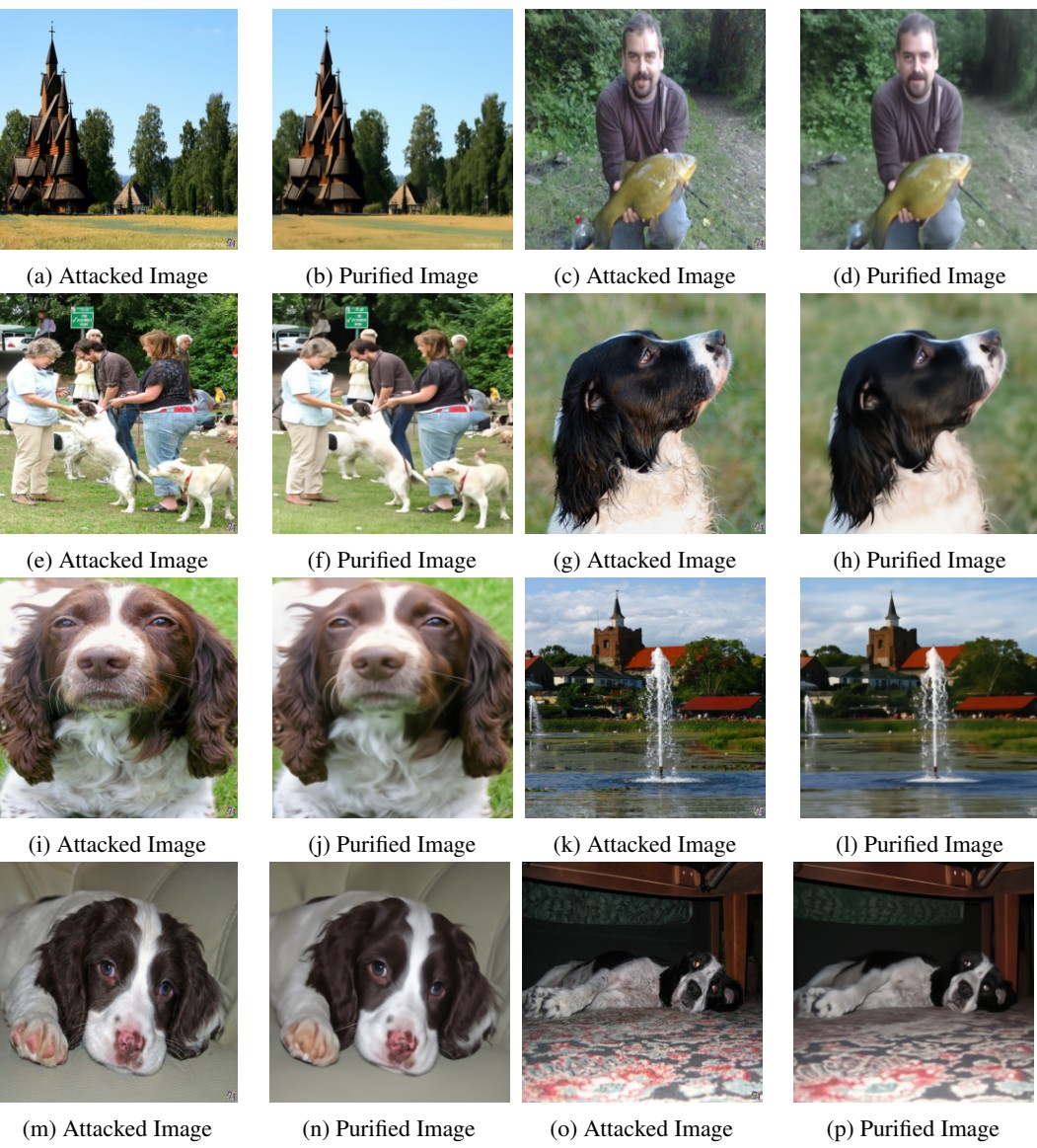

(a) Attacked Image     (b) Purified Image     (c) Attacked Image     (d) Purified Image

(e) Attacked Image     (f) Purified Image     (g) Attacked Image     (h) Purified Image

(i) Attacked Image     (j) Purified Image     (k) Attacked Image     (l) Purified Image

(m) Attacked Image     (n) Purified Image     (o) Attacked Image     (p) Purified Image

Figure 4: Comparison of Purified and BadNet Attack Images(Part1).

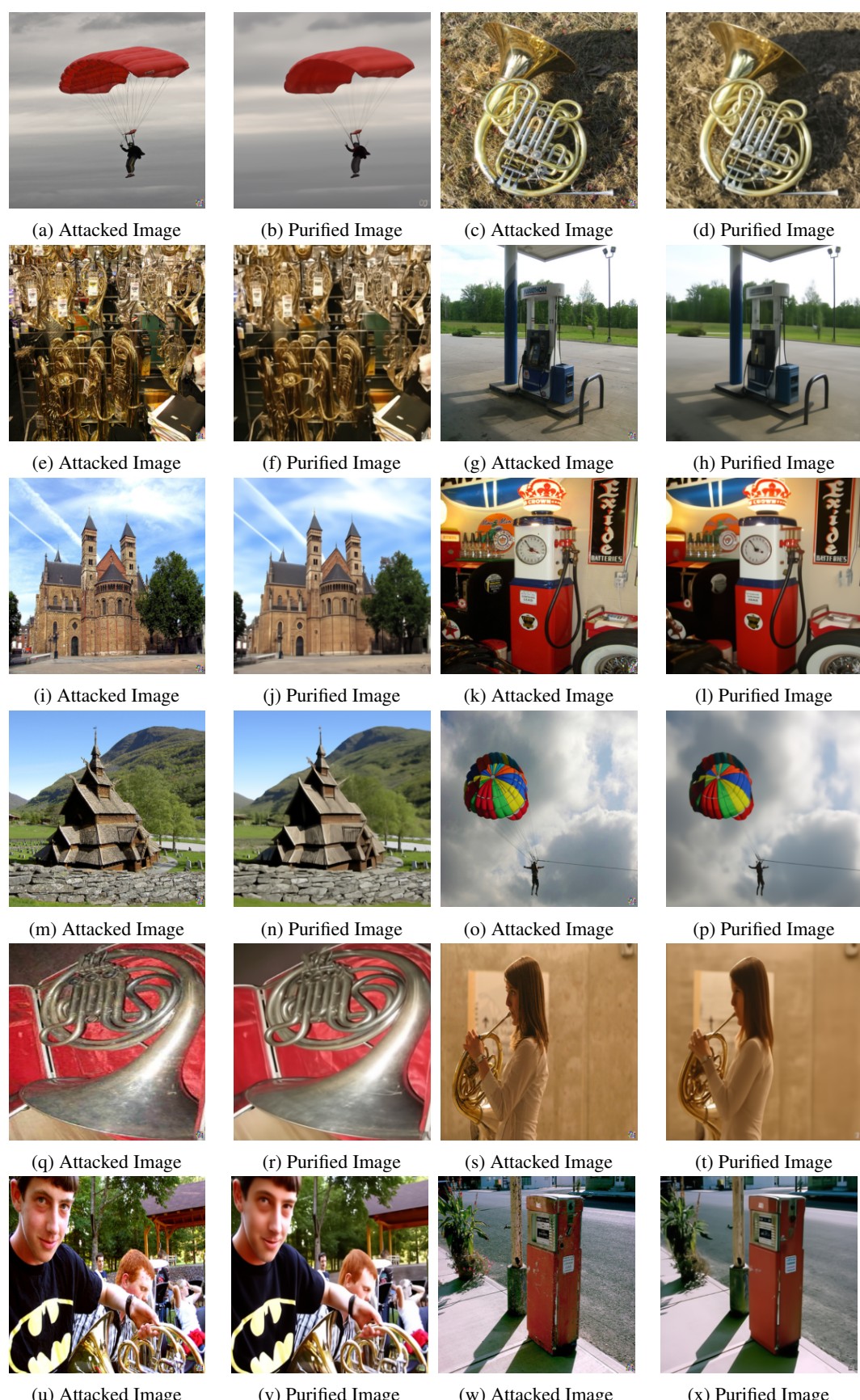

(a) Attacked Image    (b) Purified Image    (c) Attacked Image    (d) Purified Image

(e) Attacked Image    (f) Purified Image    (g) Attacked Image    (h) Purified Image

(i) Attacked Image    (j) Purified Image    (k) Attacked Image    (l) Purified Image

(m) Attacked Image    (n) Purified Image    (o) Attacked Image    (p) Purified Image

(q) Attacked Image    (r) Purified Image    (s) Attacked Image    (t) Purified Image

(u) Attacked Image    (v) Purified Image    (w) Attacked Image    (x) Purified Image

Figure 5: Comparison of Purified and BadNet Attack Images(Part2).

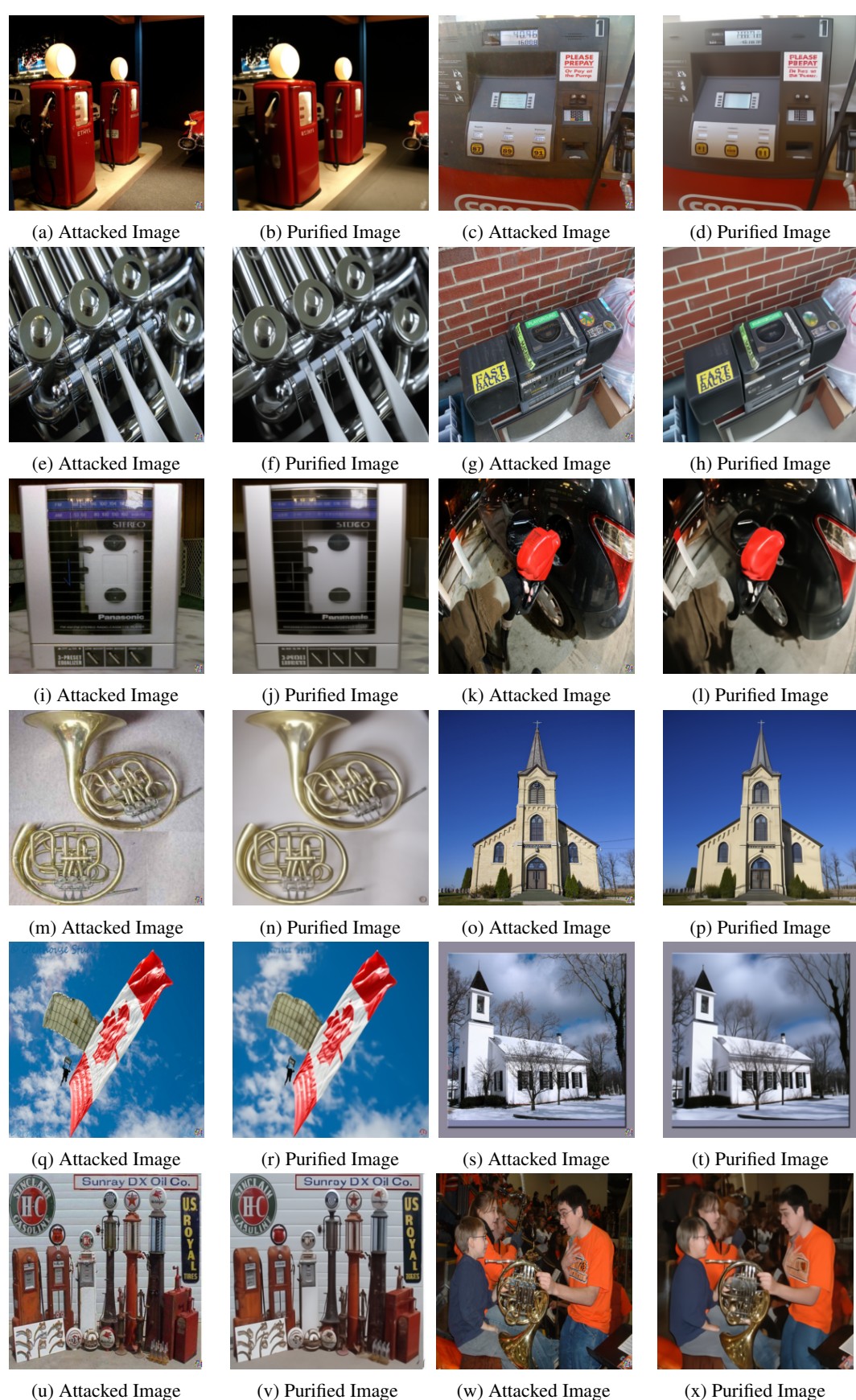

Figure 6: Comparison of Purified and BadNet Attack Images(Part3).

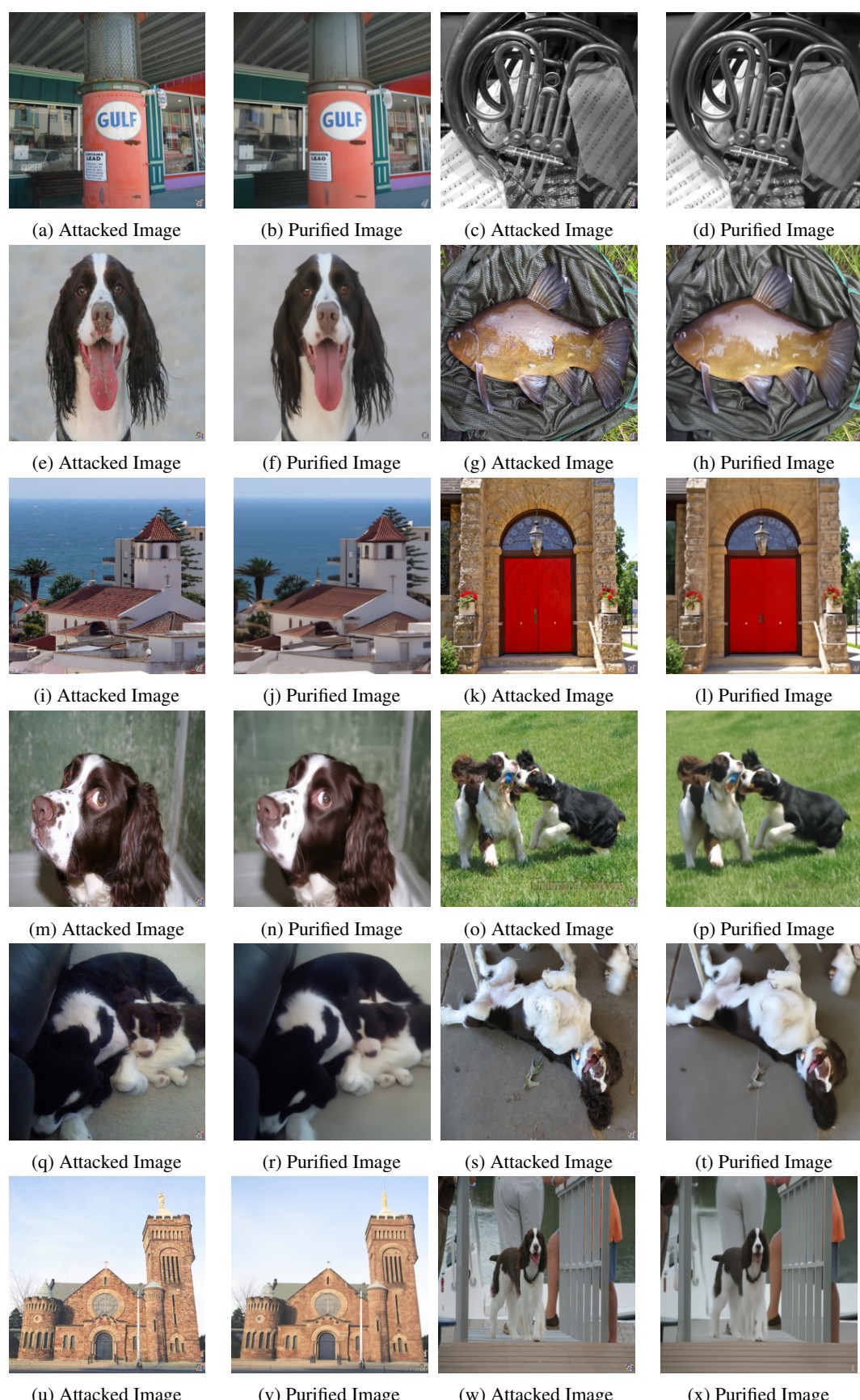

(a) Attacked Image     (b) Purified Image     (c) Attacked Image     (d) Purified Image

(e) Attacked Image     (f) Purified Image     (g) Attacked Image     (h) Purified Image

(i) Attacked Image     (j) Purified Image     (k) Attacked Image     (l) Purified Image

(m) Attacked Image     (n) Purified Image     (o) Attacked Image     (p) Purified Image

(q) Attacked Image     (r) Purified Image     (s) Attacked Image     (t) Purified Image

(u) Attacked Image     (v) Purified Image     (w) Attacked Image     (x) Purified Image

Figure 7: Comparison of Purified and BadNet Attack Images(Part4).

## B.2 Qualitative Results of Purification on Blended Attack

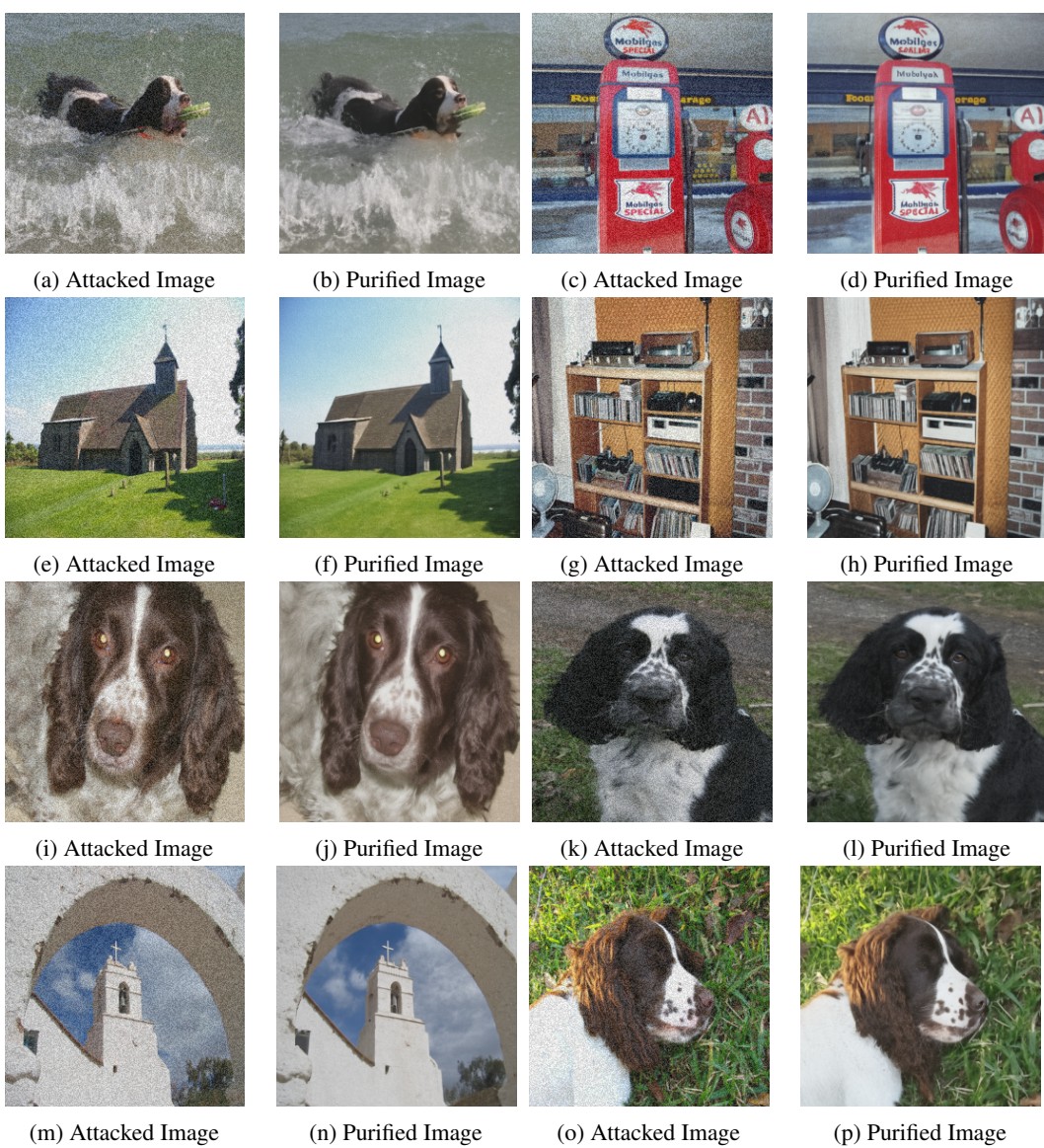

(a) Attacked Image    (b) Purified Image    (c) Attacked Image    (d) Purified Image

(e) Attacked Image    (f) Purified Image    (g) Attacked Image    (h) Purified Image

(i) Attacked Image    (j) Purified Image    (k) Attacked Image    (l) Purified Image

(m) Attacked Image    (n) Purified Image    (o) Attacked Image    (p) Purified Image

Figure 8: Comparison of Purified and Blended Attack Images(Part1).

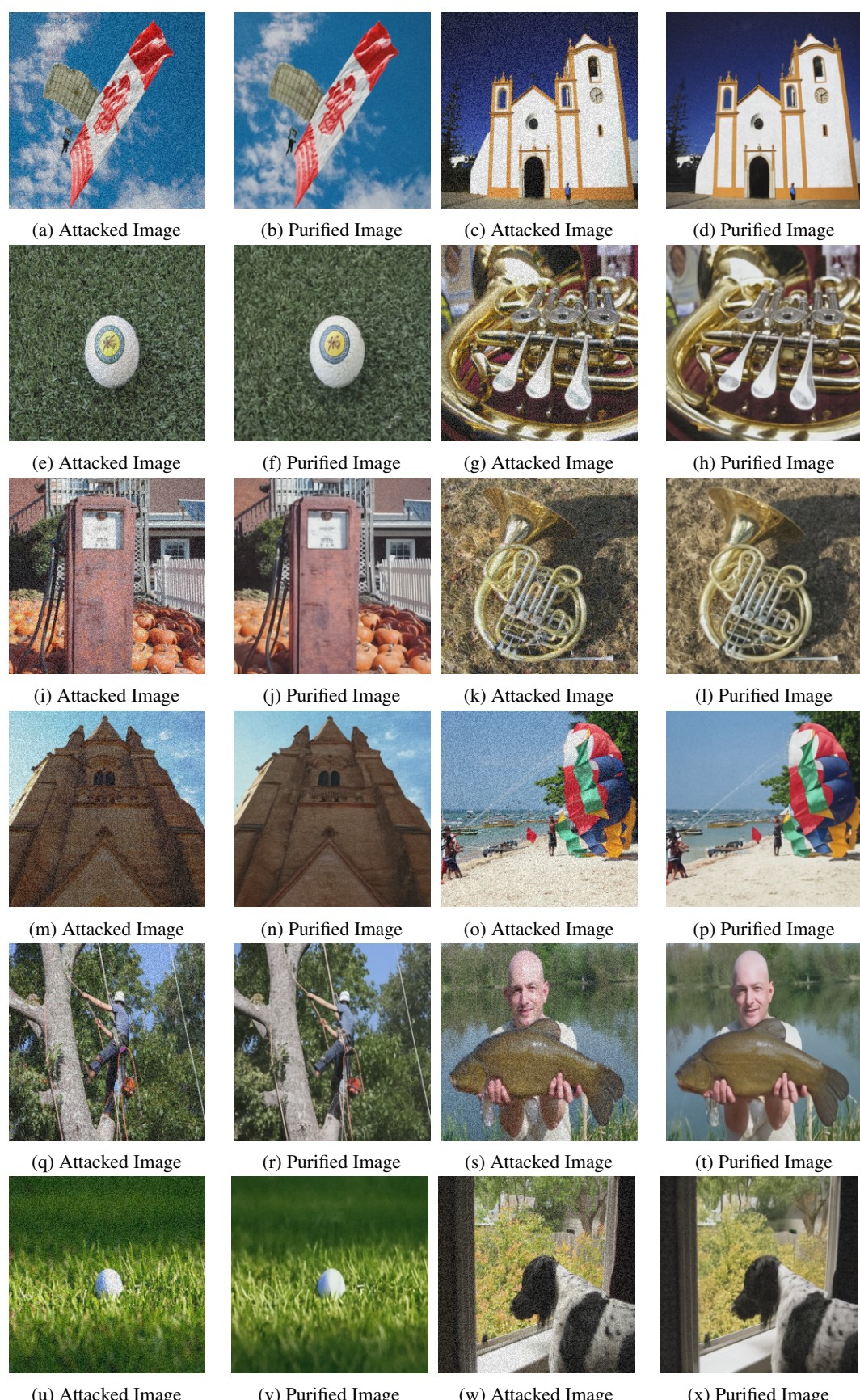

(a) Attacked Image     (b) Purified Image     (c) Attacked Image     (d) Purified Image

(e) Attacked Image     (f) Purified Image     (g) Attacked Image     (h) Purified Image

(i) Attacked Image     (j) Purified Image     (k) Attacked Image     (l) Purified Image

(m) Attacked Image     (n) Purified Image     (o) Attacked Image     (p) Purified Image

(q) Attacked Image     (r) Purified Image     (s) Attacked Image     (t) Purified Image

(u) Attacked Image     (v) Purified Image     (w) Attacked Image     (x) Purified Image

Figure 9: Comparison of Purified and Blended Attack Images(Part2).

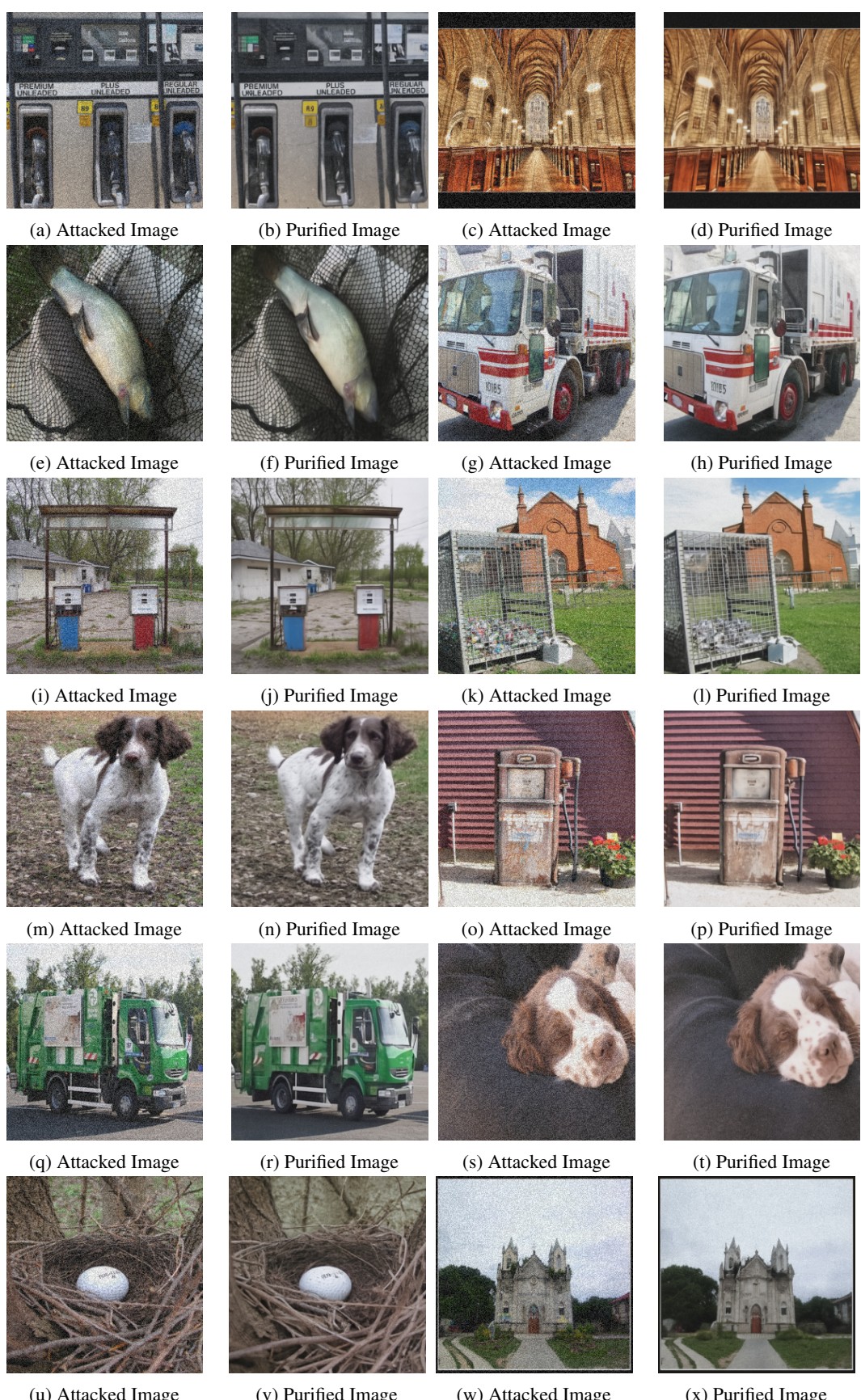

(a) Attacked Image    (b) Purified Image    (c) Attacked Image    (d) Purified Image

(e) Attacked Image    (f) Purified Image    (g) Attacked Image    (h) Purified Image

(i) Attacked Image    (j) Purified Image    (k) Attacked Image    (l) Purified Image

(m) Attacked Image    (n) Purified Image    (o) Attacked Image    (p) Purified Image

(q) Attacked Image    (r) Purified Image    (s) Attacked Image    (t) Purified Image

(u) Attacked Image    (v) Purified Image    (w) Attacked Image    (x) Purified Image

Figure 10: Comparison of Purified and Blended Attack Images(Part3).

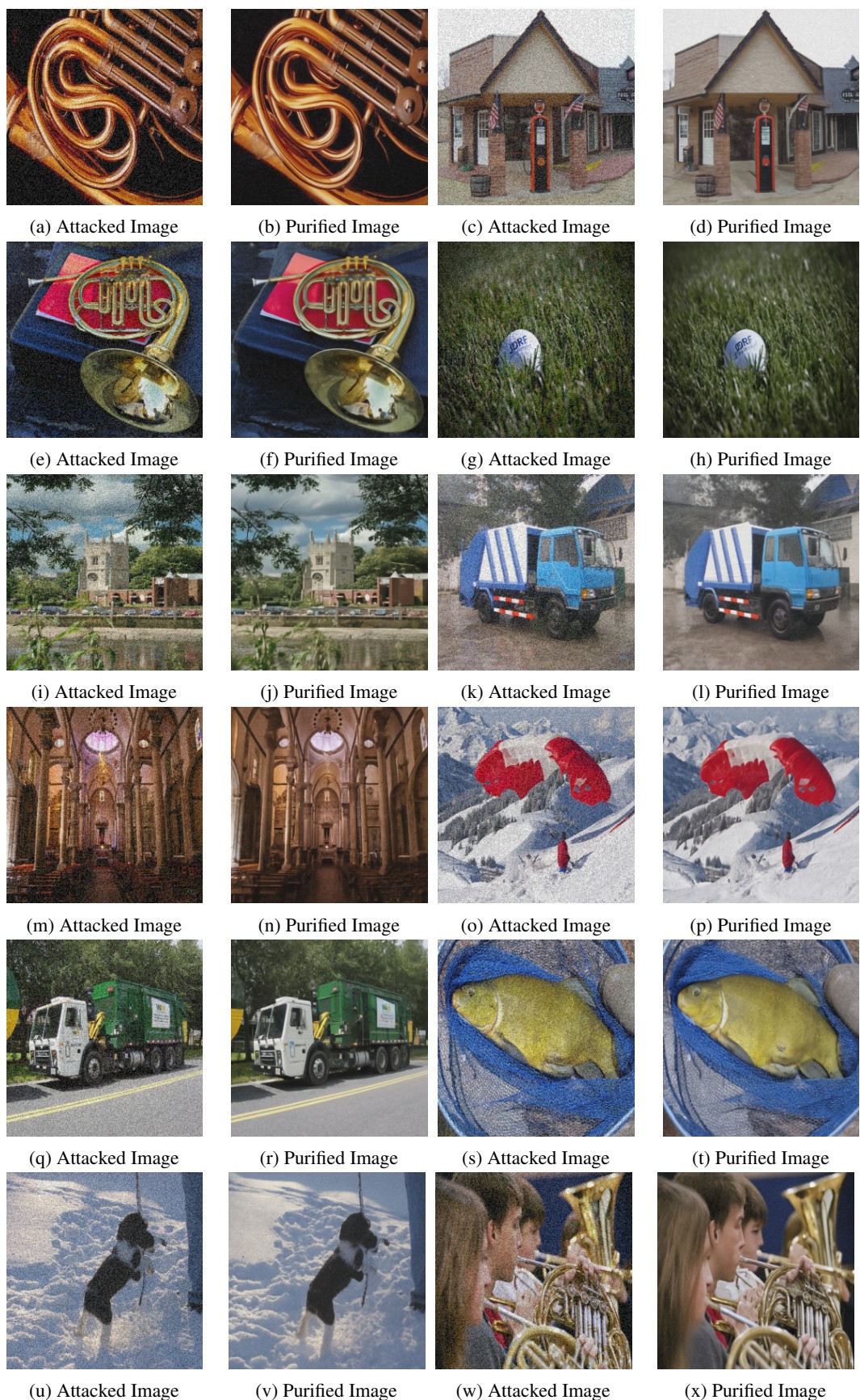

(a) Attacked Image     (b) Purified Image     (c) Attacked Image     (d) Purified Image

(e) Attacked Image     (f) Purified Image     (g) Attacked Image     (h) Purified Image

(i) Attacked Image     (j) Purified Image     (k) Attacked Image     (l) Purified Image

(m) Attacked Image     (n) Purified Image     (o) Attacked Image     (p) Purified Image

(q) Attacked Image     (r) Purified Image     (s) Attacked Image     (t) Purified Image

(u) Attacked Image     (v) Purified Image     (w) Attacked Image     (x) Purified Image

Figure 11: Comparison of Purified and Blended Attack Images(Part4).

## C   Details on Linear Transformations

In this section, we discuss details of the linear transformation applied in our paper. Two practical examples are discussed to illustrate the simplicity and effectiveness of linear transformations, along with the corresponding operators used.

**Gray-scale Conversion:** To convert an RGB image to gray-scale, the operator $\mathbf{A} = [1/3, 1/3, 1/3]$ can be defined as a pixel-wise operation that transforms each RGB channel pixel $[r, g, b]$ into a gray-scale value $r^3 + g^3 + b^3$. In this case, constructing a pseudo-inverse $\mathbf{A}^\dagger = [1, 1, 1]^\mathrm{T}$ is straightforward, satisfying the condition $\mathbf{A}\mathbf{A}^\dagger \equiv \mathbf{I}$, where $\mathbf{I}$ represents the identity matrix.

**Image Blurring:** Image blurring also involves linear transformations. For a blurring operation with scale $n$, the operator $\mathbf{A}$ is defined as the average-pooling operator $[1/n^2, ..., 1/n^2]$. This operator aggregates each patch of the image into a single value. Similarly, the pseudo-inverse $\mathbf{A}^\dagger$ can be built as $\mathbf{A}^\dagger = [1, ..., 1]^\mathrm{T}$ to fulfill the condition $\mathbf{A}\mathbf{A}^\dagger \equiv \mathbf{I}$.

Overall, these examples demonstrate how these two linear transformations, in conjunction with their respective operators, can be employed to destruct the trigger pattern without relying on a complex Fourier transform. In cases where the linear transformation is too complex to solve for its pseudo-inverse, the Singular Value Decomposition (SVD) method can be applied. For more details, please refer to papers [58, 29].

## D   Algorithm for improved ZIP based on DDIM

In this section, we include the modified algorithm based on DDIM, which is proposed to speed up the diffusion model inference speed.

---

**Algorithm 2** Zero-shot Image Purification (based on DDIM)

---

**Require:** Test image for purification $\mathbf{x}^P$; liner transformation $\mathbf{A}_1, \mathbf{A}_2, ..., \mathbf{A}_n$ and their pseudo-inverse $\mathbf{A}_1^\dagger, \mathbf{A}_2^\dagger, ..., \mathbf{A}_n^\dagger$; diffusion model $g$; hyperparameter $\lambda$; speed-up pace $S$.
**Ensure:** $\mathbf{x}^{A_n} = \mathbf{A}_n \mathbf{x}^P, \ n = 0, 1, ..., N$
1: $\mathbf{x}^T \sim \mathcal{N}(\mathbf{0}, \mathbf{I})$
2: **for** $t = T, T - S, ..., S, 1$ **do**
3:     $\boldsymbol{\epsilon} \sim \mathcal{N}(\mathbf{0}, \mathbf{I})$ if $t > 1$, else $\boldsymbol{\epsilon} = \mathbf{0}$
4:     $\boldsymbol{\epsilon}_t = g_\phi(\mathbf{x}_t, t)$
5:     **for** $n = 1, 2, ..., N$ **do**
6:         $\hat{\mathbf{x}}_t^n = \sqrt{\bar{\alpha}_t}\mathbf{A}_1^\dagger \mathbf{x}^{A_n} + (\mathbf{I} - \mathbf{A}_n^\dagger \mathbf{A}_n)\mathbf{x}_t + \mathbf{A}_n^\dagger \mathbf{A}_n \sqrt{1 - \bar{\alpha}_t}\boldsymbol{\epsilon}_t$
7:     **end for**
8:     $\hat{\mathbf{x}}_t = \frac{1}{N}(\hat{\mathbf{x}}_t^1 + \hat{\mathbf{x}}_t^2 +, ..., + \hat{\mathbf{x}}_t^N)$
9:     $\tilde{\mathbf{x}}_t = (1 - \bar{\alpha}_t^\lambda)\hat{\mathbf{x}}_t + \bar{\alpha}_t^\lambda \mathbf{x}_t$
10:     $\tilde{\mathbf{x}}_{0|t} = \frac{1}{\sqrt{\bar{\alpha}_t}}(\tilde{\mathbf{x}}_t - \sqrt{1 - \bar{\alpha}_t}\boldsymbol{\epsilon}_t)$
11:     $\mathbf{x}_{t-1} \leftarrow \sqrt{\bar{\alpha}_{t-1}}\tilde{\mathbf{x}}_{0|t} + \sqrt{1 - \bar{\alpha}_{t-1} - \sigma_t^2}\boldsymbol{\epsilon}_t + \sigma_t\boldsymbol{\epsilon}$
12: **end for**
13: **return** $\mathbf{x}_0$

---

## E   Experiments Settings

### E.1   Datasets Informaiton

**CIFAR-10 [30]** The CIFAR-10 dataset is a widely-used benchmark in computer vision. It consists of 60,000 color images of size 32x32 pixels, belonging to 10 different classes, with 6,000 images per class. The dataset is divided into 50,000 training images and 10,000 test images, with a balanced distribution of classes.

**GTSRB [52]** The German Traffic Sign Recognition Benchmark (GTSRB) dataset is designed for traffic sign classification tasks. It comprises more than 50,000 images of traffic signs captured under various real-world conditions. The images have different sizes and aspect ratios, but they are resized

to 32x32 pixels for our model training and evaluation. The dataset is divided into training and test sets, with its official split ratio.

**Imagenette [24]** The Imagenette is a subset of the larger ImageNet dataset and is commonly used as a smaller-scale alternative for image classification tasks. It consists of 10 classes with a total of 13,000 images. The images in Imagenette have varying sizes, but they are resized to 256x256 pixels for consistency. The dataset is split into training and validation sets, following a predefined split ratio.

Table 5: Properties of datasets.

| Dataset | Classes | Image Size | Train Split | Test Split |
|---------|---------|-----------|-------------|-----------|
| CIFAR-10 | 10 | 32x32 | 50,000 | 10,000 |
| GTSRB | 43 | 32x32 | 39,209 | 12,630 |
| Imagenette | 10 | 256x256 | 9,480 | 3,936 |

## E.2 Attacks Implementation

In this section, we discuss the implementation details of three different backdoor attack methods employed in our study: BadNet, Blended, and PhysicalBA. We implement these backdoor attacks using the Backdoorbox framework [33], which is under GNU general public license.

**BadNet [15]** The BadNet attack injects specific trigger patterns into the training data. In our implementation, we set the poisoned rate to 5%, i.e., 5% of the training samples are selected as attack samples and have the trigger pattern added to them. The trigger pattern size is set to 2x2 for 32x32 pixels images and 9x9 for 256x256 pixels images. The trigger patterns are randomly generated.

**Blended [8]** The Blended attack is a more sophisticated variant aimed at making the backdoor less conspicuous and harder to detect. Following the suggestion in BackdoorBox, we set the blended rate to 0.2 and the poisoned rate to 5%. The blended pattern is randomly generated, seamlessly blending the trigger pattern into the attack samples.

**PhysicalBA [34]** The PhysicalBA (Physical Backdoor Attack) is a specific type of attack that introduces variations in the location and appearance of the attack pattern embedded in the test samples during inference time. In our implementation, we apply the same attack pattern size as the BadNet attack, using a 2x2 pattern for 32x32 pixels images and a 9x9 pattern for 256x256 pixels images. The attack patterns are generated randomly. We set the poisoned rate to 5% for this attack.

All other attack settings follow the default configurations in Backboorbox [33].

## E.3 Purification Implementation

We utilize a pre-trained model provided by OpenAI [9] under the MIT license. The algorithm described in Algorithm 2 is employed to accelerate the inference process, allowing us to generate high-quality images within just 20 steps, and the speed-up pace is set to 50. We set the hyperparameter $\lambda$ to a value of 2 for Blended attack defense, and 10 for BadNet and PhysicalBA attack defense.

Specifically, for the CIFAR-10 dataset, we apply both blur and gray-scale conversion as linear transformations. For the GTSRB and Imagenette datasets, we solely apply blur as the linear transformation. Additional implementation details can be found in the code we have provided.

# F   Ablation Study Settings

## F.1 Enhanced Attack Settings

In the enhanced attack settings, our first step is to extract 5% of the training dataset and inject the attack's trigger pattern into these images. We then proceed to purify this subset of data using blur and grayscale as linear transformations during the first stage of our proposed purification. Once the attacked images have been successfully purified, we modify their labels to reflect the attack label. Following this, we introduce these purified images as poisoned samples into the training set and train a classification model from scratch. This comprehensive procedure is referred to as the enhanced attack process.

## F.2 Purification Speed Settings

This paper focuses on defending against backdoor attacks during the inference phase using purification techniques. To evaluate the purification speed, we conduct experiments using a workstation that features an Intel(R) Core(TM) i9-10900X CPU and an NVIDIA RTX3070 GPU with 8GB of memory.

During our experiments, we measure the classification time, which represents the duration taken by the classifier model to perform inference on a single image. Additionally, we measure the purification time, which indicates the time required by the purification model to purify a single image.

For the combination of purification with detection, we utilize the Scale-up method [19] as our chosen detection technique. Furthermore, the dataset used for speed evaluation consists of 5% poisoned images. Following previous settings [19], we set a batch size of one for the classifier model and report the average time based on 640 runs.

# G   Related Work

## G.1   Backdoor Attack

Existing backdoor attack methods involve the injection of poisoned samples into the training process of Deep Neural Networks (DNNs). These attacks can target various types of models, including image classification models, object detection models [5, 36], contrastive learning models [3], and language models [40, 49, 69, 6]. The attackers exploit vulnerabilities by embedding adversary-specified trigger patterns into carefully selected benign samples. Backdoor attacks are characterized by their stealthiness, as the attacked models behave normally on benign samples, making the hidden triggers difficult to detect and purify.

There are mainly two categories of backdoor attacks for image classification tasks: patch-based and non-patch-based attacks. Patch-based attacks are attacks with triggers embedded as patches or overlays within the input samples. For example, Souri et al. [51] propose the Sleeper Agent attack, which is a sophisticated backdoor attack where an adversary subtly injects hidden triggers into an image classification model during training, remaining dormant until specific conditions activate malicious behavior. Non-patch-based attacks are attacks where triggers are integrated without explicit patching, often relying on specific input sequences or subtle modifications in the feature space [67, 23, 20]. For example, Doan et al. [11] introduce Wasserstein backdoor attack, an extension of the imperceptible backdoor concept to the latent representation. Their proposed attack manipulates inputs with imperceptible noise, matching latent representations to achieve high attack success rates while remaining stealthy in both the input and latent spaces.

## G.2   Backdoor Defense

Existing defense methods for backdoor models can be broadly categorized into two approaches: (1) detection-based methods and (2) purification-based methods.

Detection-based methods focus on identifying the presence of backdoors in trained models. These methods typically involve analyzing the model's behavior and inputs to detect any suspicious patterns or triggers that indicate the existence of a backdoor [2]. Various techniques such as anomaly detection [13, 25, 63], and statistical analysis [18, 7] have been employed to detect backdoors. The goal of detection-based methods is to provide an early warning system to identify and mitigate the risks posed by backdoor attacks.

On the other hand, purification-based methods aim to remove or neutralize the effects of backdoors from the model. These methods involve modifying the model or its training process to eliminate the influence of the backdoor triggers on the model's behavior [26, 39]. Some purification approaches focus on retraining the model using clean or carefully selected training data to reduce the impact of the backdoor [64, 59, 31, 27, 55]. Other methods aim to directly identify and neutralize the backdoor triggers within the model's parameters or hidden representations [57, 4, 62, 60, 41]. The objective of purification-based methods is to restore the integrity and reliability of the model by eliminating the malicious behavior induced by the backdoor.

# H  Comparison Results with Image Restoration/Purification Methods

In this section, we compare our proposed ZIP with existing purification and restoration methods, including DiffPure [44] and DDNM [58]. Specifically, DiffPure is a state-of-the-art purification method designed for adversarial attacks, while DDNM is a state-of-the-art image restoration technique to repair corrupted images.

## H.1  Quantitative Results Comparison with Image Restoration/Purification Methods

We first conduct a quantitative analysis of the defense performance between our proposed method and DiffPure and DDNM. In order to ensure a fair comparison, we implemented DiffPure and DDNM using their official code and applied identical linear transformations, diffusion steps, and schedules to both methods. The defense performance is listed in Table 6.

Table 6: Defense performance on Imagenette.

| Attack | No Defense | | | DiffPure | | | Blur+DiffPure | | | Blur+DDNM | | | ZIP (Ours) | | |
|---|---|---|---|---|---|---|---|---|---|---|---|---|---|---|---|
| | CA | ASR | PA | CA | ASR | PA | CA | ASR | PA | CA | ASR | PA | CA | ASR | PA |
| BadNet | 84.99 | 94.53 | 14.98 | 78.85 | 91.41 | 18.11 | 75.13 | 10.16 | 74.31 | 84.56 | 9.14 | 82.24 | 84.05 | 7.55 | 83.97 |
| Blended | 86.14 | 99.85 | 10.19 | 80.63 | 43.82 | 56.68 | 75.89 | 13.53 | 73.98 | 85.37 | 93.37 | 15.46 | 81.42 | 8.35 | 78.36 |

We can first observe that our ZIP performs better than DiffPure and Blur+DiffPure in all three metrics regarding defense performance. This is because our ZIP can recover the semantic information removed by the Blur, while Blur+DiffPure can not. Specifically, DiffPure aims to remove attack patterns by first diffusing images with noise and then recovering images through an unconditional reverse process. While coupling DiffPure with Blur enhances its trigger removal capability (lower ASR), its unconditional generative process fails to effectively recover the semantic information removed by Blur. This leads to a drop in clean accuracy (CA). In the above table, Blur+DiffPure exhibits poorer CA compared to both DiffPure and our ZIP. In contrast, our ZIP, after Blur, utilizes a conditional reverse process to recover semantic information deleted by Blur through RND theory. Therefore, our ZIPs can outperform DiffPure.

We can also observe that DDNM shows better clean accuracy than ZIP, but it cannot effectively defend against backdoor attacks like Blended. This is because DDNM restores the attack patterns in its diffusion process, which decreases the defense performance.

## H.2  Qualitative Results Comparison with Image Restoration Methods

In this subsection, we conduct a qualitative analysis of the purification effect between our proposed method and DDNM.

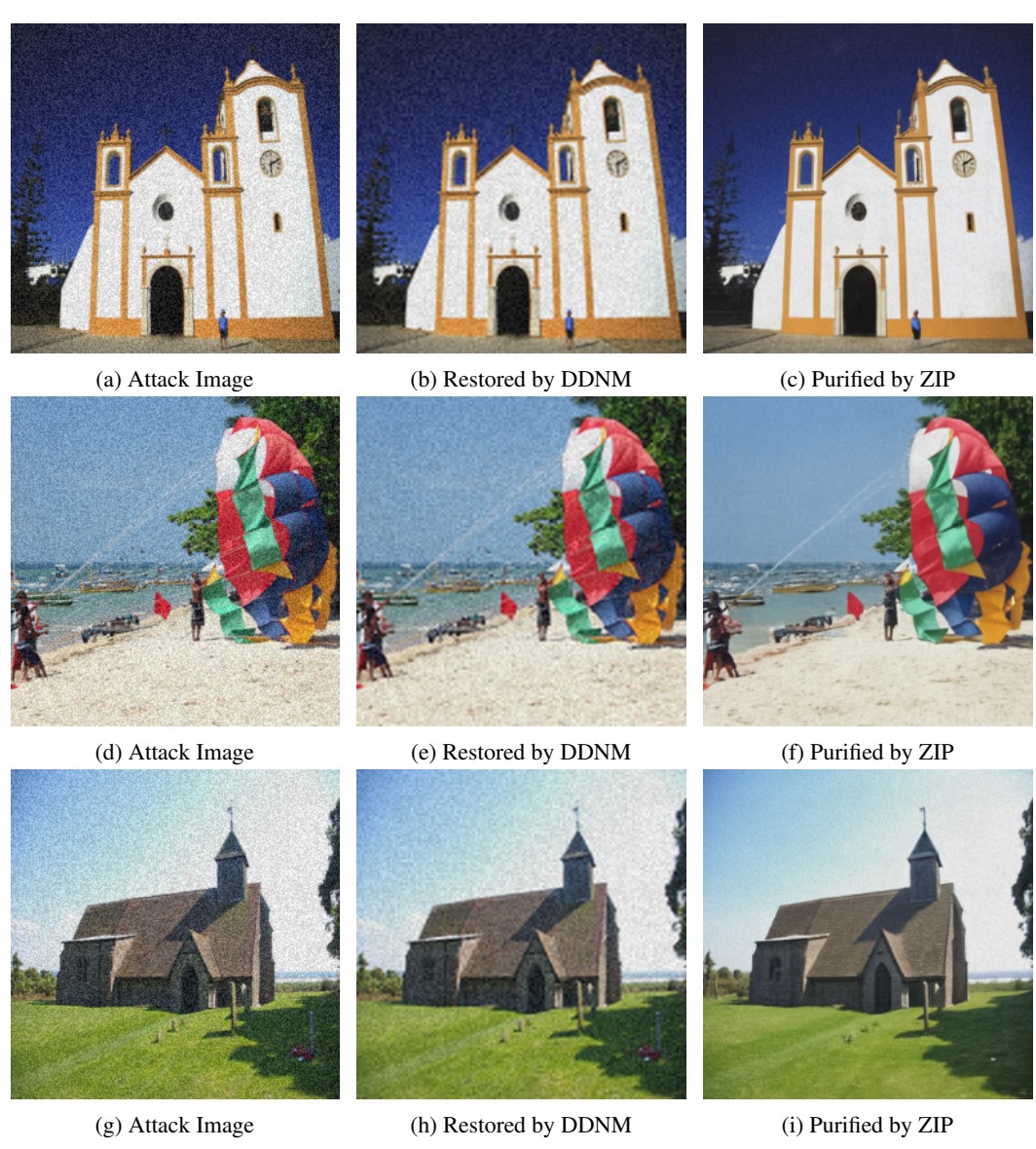

(a) Attack Image        (b) Restored by DDNM        (c) Purified by ZIP

(d) Attack Image        (e) Restored by DDNM        (f) Purified by ZIP

(g) Attack Image        (h) Restored by DDNM        (i) Purified by ZIP

Figure 12: Comparison of DDNM and ZIP on defending Blended attack.

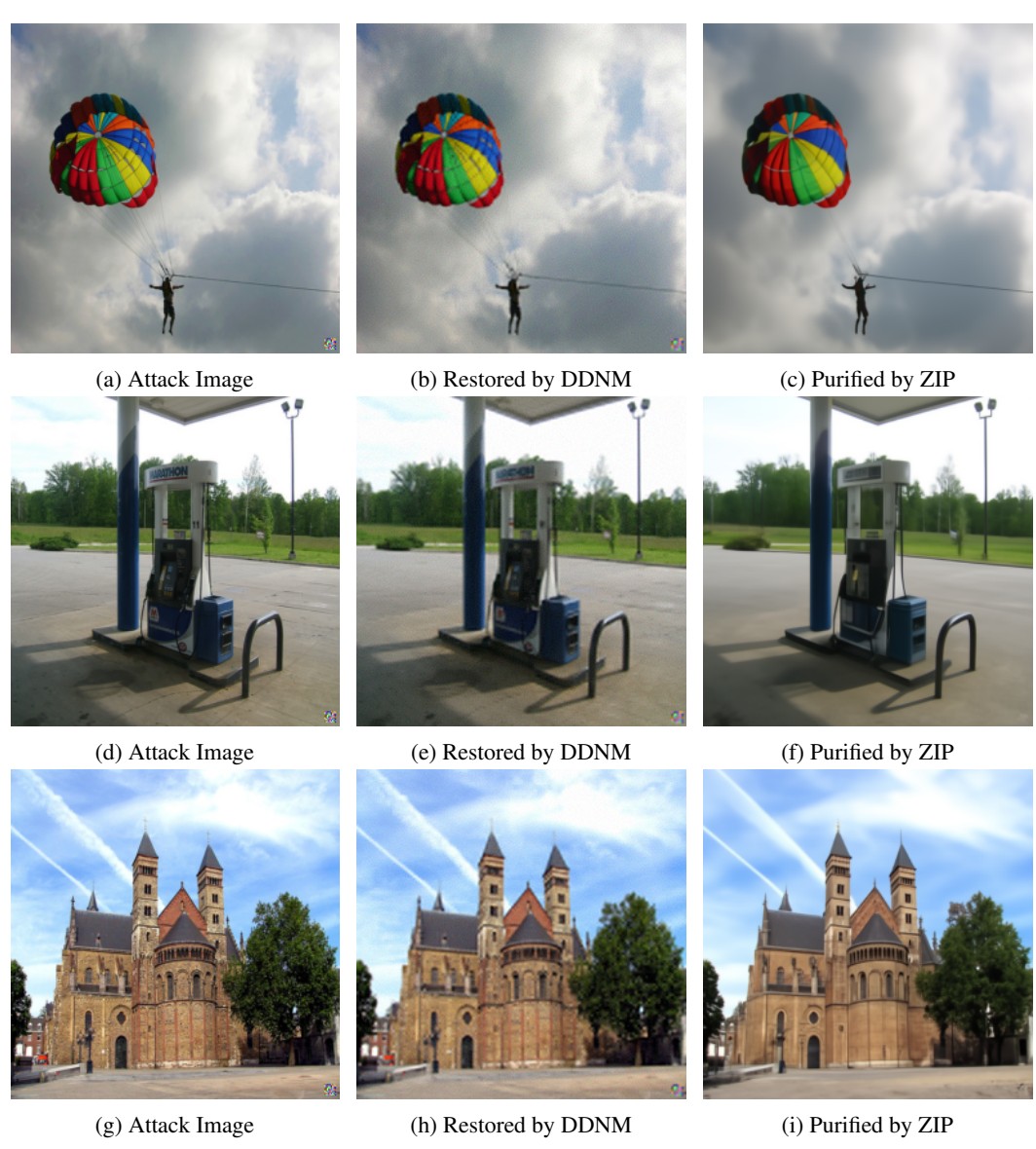

(a) Attack Image     (b) Restored by DDNM     (c) Purified by ZIP

(d) Attack Image     (e) Restored by DDNM     (f) Purified by ZIP

(g) Attack Image     (h) Restored by DDNM     (i) Purified by ZIP

Figure 13: Comparison of DDNM and ZIP on defending BadNet attack.

# I   Limitations

Due to our reliance on a pre-trained diffusion model to implement zero-shot purification, the effectiveness of generating purified images may be weakened when our model is applied to highly specific images that fall outside the distribution of pre-processed data. To mitigate this issue, we suggest two possible solutions in future work: 1) replacing the current pre-trained diffusion model with a more suitable pre-trained model for such specific images, and 2) collecting a subset of highly specific images to perform fine-tuning on the pre-trained model.

