# OpenReview forum: "Black-box Backdoor Defense via Zero-shot Image Purification"
_NeurIPS.cc/2023/Conference — NeurIPS 2023 poster_

### Official Review · Reviewer_8ejw · 2023-07-05

**Soundness:** 3 good
**Presentation:** 3 good
**Contribution:** 3 good
**Rating:** 5
**Confidence:** 5

**Summary:**

In this paper, authors proposed a two-stage framework. Based on Range-Null Space Decomposition theory, they first utilize image transformation to destruct the trigger pattern, and then leverage a pre-trained diffusion model to restore the semantic information.

**Strengths:**

(1)	To the best of my knowledge, authors propose a novel black-box backdoor defense method in a zero-shot setting.
(2)	The contributions mentioned by the authors in this paper are reasonable.


**Weaknesses:**

The chosen baselines of experiments in Section 4.3 (the results in Table 1) are not convincing and sufficient.

**Questions:**

What is the motivation of introducing RND?  Without RND, is there any change of the results? As shown in this paper(line 261), “Defense methods available for black-box purification in zero-shot are rare”, why not setting the black-box backdoor defense Salient Conditional Diffusion (Sancdifi) as your baseline?

**Limitations:**

Line 52-54 of this paper, If there is evidence in other existing literature to support this sentence “the semantic information in a poisoned image (e.g., faces, cars, or buildings) constitutes the majority of the data” , please cite it; if not, this idea should be used as a methodological assumption.

---

> ### Author Rebuttal · Authors · 2023-08-10
>
> ***W1&Q2:The chosen baselines of experiments in Section 4.3 (the results in Table 1) are not convincing and sufficient. As shown in this paper(line 261), “Defense methods available for black-box purification in zero-shot are rare”, why not setting the black-box backdoor defense Salient Conditional Diffusion (Sancdifi) as your baseline?***
>
> Thank you for the suggestion. To our best knowledge, there is no publicly available code for Salient Conditional Diffusion (Sancdifi) for replication. In our best effort to include recent baseline methods, **we include contemporaneous work BDMAE[1] in Section 4.4.1**, using its available online code. BDMAE is a black-box defense method that first identifies and masks trigger regions, then restores them using a masked autoencoder.
>
> As shown in Table 3, BDMAE can defend against patch-based attacks like BadNets (ASR = 1.12), but it fails in non-patch attacks like Blended (ASR = 99.88). In contrast, **our ZIP method can effectively defend against both kinds of attacks,** making it a more practical defense against various types of attacks.
>
> It is also worth clarifying that black-box purification in the zero-shot setting is very challenging, and as a result, baseline methods are indeed rare.
>
> [1] Sun, Tao, et al. "Mask and Restore: Blind Backdoor Defense at Test Time with Masked Autoencoder."  arXiv, 2023
>
>
>
> ***Q1: What is the motivation of introducing RND? Without RND, is there any change of the results?***
>
> The motivation of introducing the RND theory is to **guide the diffusion model in generating high-fidelity purified images.** Without RND, the reverse diffusion process becomes uncontrollable, resulting in random purified images with unpredictable semantic information. Corollary 3.2.1, presented in Section 3.3.2, theoretically validates the rationale for incorporating RND into our approach.
>
> ***L1: Line 52-54 of this paper, If there is evidence in other existing literature to support this sentence “the semantic information in a poisoned image (e.g., faces, cars, or buildings) constitutes the majority of the data” , please cite it; if not, this idea should be used as a methodological assumption.***
>
> Thanks for the suggestion. This sentence can be supported by  [1,2,3]. We will add these references In our next version.
>
> [1] Li, Yiming, et al. "Backdoor learning: A survey."  TNNLS, 2022
> [2]  Li, Yiming, et al. "Backdoor attack in the physical world." ICLR workshop, 2021.
> [3] Sun, Tao, et al. "Mask and Restore: Blind Backdoor Defense at Test Time with Masked Autoencoder."  arXiv, 2023

---

### Official Review · Reviewer_8w5v · 2023-07-07

**Soundness:** 3 good
**Presentation:** 3 good
**Contribution:** 3 good
**Rating:** 5
**Confidence:** 4

**Summary:**

The paper proposes a backdoor defense for real-world black-box models through zero-shot image purification (ZIP). The proposed defense framework includes two stages: (1) applying linear transformation on a poisoned sample (2) using a pre-trained diffusion model to recover semantic information removed by stage (1). The proposed ZIP is utilized against different backdoor attacks on different datasets. The experiments demonstrate the effectiveness of proposed ZIP.

**Strengths:**

The proposed idea is interesting and the authors analyze the zero-shot image purification theoretically. The conducted experiments demonstrate the effectiveness of proposed method.

**Weaknesses:**

1. The authors only use three backdoor attacks including BadNet, PhysicalBA and Blended. Please use more attacks such as WaNet, label-clean attacks to verify the effectiveness of proposed method.
2. In the Introduction Section (line 31 - 36), the authors mention that there are some works about black-box setting. Please compare with these methods.
3. The paper needs to be proofread. There are some typos in the paper e.g. in Algorithm 1 (line 222) "liner" should be "linear".

**Questions:**

1. The proposed method uses a pre-trained diffusion model to recover semantic. Does the diffusion model need addition in-distribution clean and poisoned data to train or fine-tune?
2. Could the proposed method work for different network architectures?
3. In Table 3, BDMAE gets a higher CA than ZIP. Could the authors explain this?
4. If the input is a clean image, is there a possibility that ZIP can misclassify the clean image? Could the authors provide some failure samples?

**Limitations:**

The authors have addressed the limitations

---

> ### Author Rebuttal · Authors · 2023-08-10
>
> ***W1: The authors only use three backdoor attacks including BadNet, PhysicalBA and Blended. Please use more attacks such as WaNet, label-clean attacks to verify the effectiveness of proposed method.***
>
> Thanks for the suggestion. We conduct further experiments with the **WaNet[1], Blind[2], Label-Consistence[3]** attack on Imagenette datasets to further demonstrate our defense effectiveness. The quantitative results are listed below, while the qualitative results are provided in the rebuttal PDF. The results show that our ZIP can successfully defend against various advanced attacks.
>
> |  | No Defense         | ShrinkPad (defense) | Blur (defense)    | ZIP (Ours)           |
> | ---------------------- | ------------------ | ------------------- | ----------------- | -------------------- |
> | Imagenette (256 × 256) | CA ↑/ ASR ↓/ PA ↑  | CA ↑/ ASR ↓/ PA ↑   | CA ↑/ ASR ↓/ PA ↑ | CA ↑/ ASR ↓/ PA ↑    |
> | WaNet                  | 85.45 /97.93/11.84 | 73.63 /26.87/60.35  | 62.98/17.88/56.81 | **79.13/7.77/76.78** |
> | Blind                  | 79.15/99.64/10.42  | 65.22/6.34/65.60    | 75.26/28.12/67.54 | **78.80/5.32/79.03** |
> | LabelConsistent        | 77.57/73.17/36.56  | 66.14/0.25/64.53    | 50.36/0.28/50.14  | **73.63/0.12/74.26** |
>
> [1]Wanet--imperceptible warping-based backdoor attack. ICLR, 2021.
> [2]Blind backdoors in deep learning models. USENIX Security, 2021.
> [3]Label-consistent backdoor attacks. arXiv 2019.
>
> ***W2: Please compare with works under black-box setting.***
>
> Thanks for this comment. We provide a comprehensive comparison in **Supplementary Material G.** In brief, the existing defense methods in black-box settings are mainly constrained to:
> 1. **Backdoor Detections**: Identifying backdoors in poisoned samples [1, 2, 3, 4], without mitigating the poisoning effect.
> 2. **Limited Purification**: Removing backdoors patterns under specific circumstances, such as partially white-box settings [5], or focusing solely on patch-based attacks [6].
>
> Compared to existing work, our ZIP can achieve **zero-shot** backdoor **purification** encompassing various attack patterns (patch-based and non-patch-based) under the **black-box setting**. These features make it a unique and powerful defense that overcomes the limitations of existing methods.
>
> [1]DeepInspect: A Black-box Trojan Detection and Mitigation Framework for Deep Neural Networks. IJCAI, 2019.
> [2]Aeva: Black-box backdoor detection using adversarial extreme value analysis.ICLR, 2022.
> [3]Scale-up: An efficient black-box input-level backdoor detection via analyzing scaled prediction consistency.ICLR, 2023.
> [4]Black-box detection of backdoor attacks with limited information and data.ICCV, 2021.
> [5]Februus: Input purification defense against trojan attacks on deep neural network systems. ACSAC, 2020.
> [6]Mask and Restore: Blind Backdoor Defense at Test Time with Masked Autoencoder.arXiv, 2023.
>
> ***W3:The paper needs to be proofread. There are some typos in the paper e.g. in Algorithm 1 (line 222) "liner" should be "linear".***
>
> Thank you for bringing this to our attention. We will correct these typos in our next version.
>
> ***Q1: The proposed method uses a pre-trained diffusion model to recover semantic. Does the diffusion model need addition in-distribution clean and poisoned data to train or fine-tune?***
>
> Our framework does not require additional in-distribution of clean or poisoned data to train or fine-tune. In our current experiments, we choose a diffusion model [1] pre-trained on ImageNet as our backbone model. The selected model demonstrates great zero-shot purification performance on both in-distribution (Imagenette) and out-of-distribution (CIFAR-10, GTSRB) images.
>
> [1]Diffusion models beat gans on image synthesis.NeurIPS, 2021.
>
> ***Q2: Could the proposed method work for different network architectures?***
>
> Thank you for this comment. Yes, it can work for classifiers with different network architectures. To verify this, we implemented ZIP with a new classifier using VGG-16 Network. The defense performance with VGG-16 and ResNet-34 are listed as follows:
>
>
> |  | No defense with ResNet-34 | ZIP with ResNet-34 | No defense with VGG-16 | ZIP with VGG-16   |
> | ---------------------- | ------------------------- | ------------------ | ---------------------- | ----------------- |
> | Imagenette (256 × 256) | CA ↑/ ASR ↓/ PA ↑ | CA ↑/ ASR ↓/ PA ↑  | CA ↑/ ASR ↓/ PA ↑| CA ↑/ ASR ↓/ PA ↑ |
> | BadNet  | 84.99/94.53/14.98         | 84.05/7.55/83.97   | 70.24/91.67/17.35      | 68.71/9.72/69.09  |
> | Blended | 86.14/99.85/10.19         | 81.42/8.35/78.36   | 75.33/96.17/12.76      | 74.16/6.31/70.77  |
> | PhysicalBA | 90.67/72.94/34.29 | 87.26/10.91/86.54  | 88.89/99.54/10.49      | 85.09/11.53/83.10 |
>
> ***Q3: In Table 3, BDMAE gets a higher CA than ZIP. Could the authors explain this?***
>
> Thank you for the thoughtful comment. BDMAE achieves a higher CA by applying smaller masking perturbation, which better preserves semantic information in the purified image. BDMAE first identifies and masks trigger regions, then restores them using a masked autoencoder. The smaller masked regions introduce less perturbation compared to the linear transformation (e.g., Blur) in our method.
>
> We would also like to point out that, as shown in Table 3, BDMAE cannot defend against non-patch attacks like Blended (ASR after BDMAE is 99.88). In contrast, our ZIP method can effectively defend against various attacks.
>
> ***Q4: If the input is a clean image, is there a possibility that ZIP can misclassify the clean image? Could the authors provide some failure samples?***
>
> It is possible but the probability is low. The experiments in Table 1 show that after the purification, the classification accuracy on clean images (CA) experiences a relatively marginal average drop of 3.09% across all three datasets, showing its effectiveness in maintaining information of clean data.

---

> > ### Comment · Reviewer_8w5v · 2023-08-15
> >
> > I have read the rebuttal, and most of my concerns are solved. Thanks!

---

### Official Review · Reviewer_xL3D · 2023-07-12

**Soundness:** 3 good
**Presentation:** 3 good
**Contribution:** 2 fair
**Rating:** 5
**Confidence:** 3

**Summary:**

This work proposes backdoor defense for scenarios in which a defender does not need to access the internal model, a.k.a., the black-box backdoor defense. Firstly, a linear image transformation is used to destroy potential trigger patterns. Then, a pre-trained diffusion model is used to reconstruct the missing information induced in the linear transformation procedure, where a reverse process was designed to satisfy the conditional image generation/reconstruction. Defense experiments were conducted against three backdoor attacks to show its defensive effects.

**Strengths:**

1.	The paper writes well and is easy to follow.
2.	The reported results indicate the effectiveness of the proposed method.
3.	Instead of directly applying diffusion model, adaptive reverse process was proposed.


**Weaknesses:**

1.	Though it could find certain applications, the concept of black-box defense does not seem very practical and hence not quite attractive to me.
2.	For the image reconstruction stage, there already exist a vast amount of advanced image restoration methods, such as diffusion model based e.g. DDNM in [1] or Restormer in [2]. I am wondering how would these pretrained models work to replace the reconstruction process in the second stage.
3.	The authors used zero-shot image purification, however, it depends on a pretrained diffusion model. Have the authors evaluated how would the performance change if using a different diffusion model, or training on a different dataset?
3.	The only baseline method was from 2020, which is too old in my opinion. More recent baselines should be evaluated.

References

[1] Wang, Yinhuai, Jiwen Yu, and Jian Zhang. "Zero-shot image restoration using denoising diffusion null-space model." arXiv preprint arXiv:2212.00490 (2022).

[2] Zamir, Syed Waqas, et al. "Restormer: Efficient transformer for high-resolution image restoration." Proceedings of the IEEE/CVF conference on computer vision and pattern recognition. 2022.

**Questions:**

I listed the questions in Weakness.

**Limitations:**

Yes

---

> ### Author Rebuttal · Authors · 2023-08-10
>
> **W1: The concept of black-box defense does not seem very practical and hence not quite attractive to me.**
>
> We appreciate your concerns and wish to highlight that **the black-box defense setting has been widely explored[1, 2, 3, 4]**. In applications like fraud detection, organizations (e.g., financial institutions) often purchase machine learning services from vendors or directly exploit third-party DNNs downloaded online. These systems could harbor backdoors introduced by malicious code or data poisoning[5, 6]. Due to intellectual property concerns, these systems are typically black-box to end-users, restricting access only to query-based APIs, thus posing challenges to end-users effective defense strategies. In these scenarios, our ZIP model can effectively serve as a "firewall," blocking and purifying malicious samples.
>
> In addition to its black-box capabilities, our framework is also **model-agnostic** and **zero-shot**. “Model-agnostic” means our framework enables seamless adaptation to new downstream classifiers without retraining, as demonstrated with VGG-16 and ResNet-34 below. “Zero-shot” means our ZIP does not require any prior knowledge of poisoned images, making it more practical since new attacks always emerge while defenders may not have access to new attack samples.
>
> |                        | No defense (ResNet-34) | ZIP (ResNet-34) | No defense (VGG-16) | ZIP (VGG-16 )  |
> | ---------------------- | ------------------------- | ------------------ | ---------------------- | ----------------- |
> | Imagenette (256 × 256) | CA ↑/ ASR ↓/ PA ↑         | CA ↑/ ASR ↓/ PA ↑  | CA ↑/ ASR ↓/ PA ↑      | CA ↑/ ASR ↓/ PA ↑ |
> | BadNet                 | 84.99/94.53/14.98         | 84.05/7.55/83.97   | 70.24/91.67/17.35      | 68.71/9.72/69.09  |
> | Blended                  | 86.14/99.85/10.19         | 81.42/8.35/78.36   | 75.33/96.17/12.76      | 74.16/6.31/70.77  |
> | PhysicalBA             | 90.67/72.94/34.29         | 87.26/10.91/86.54  | 88.89/99.54/10.49      | 85.09/11.53/83.10 |
>
> [1] DeepInspect: A Black-box Trojan Detection and Mitigation Framework for Deep Neural Networks. IJCAI, 2019.
> [2] Aeva: Black-box backdoor detection using adversarial extreme value analysis. ICLR, 2022.
> [3] Scale-up: An efficient black-box input-level backdoor detection via analyzing scaled prediction consistency.ICLR, 2023.
> [4] Black-box detection of backdoor attacks with limited information and data.ICCV, 2021.
> [5] Badnets: Identifying vulnerabilities in the machine learning model supply chain.arXiv, 2017.
> [6] Blind backdoors in deep learning models.USENIX Security, 2021.
>
> ***W2: Evaluation of image restoration methods, such as diffusion model based e.g. DDNM in [1] or Restormer in [2].***
>
> We have included the **qualitative** performance comparison with DDNM in **Supplementary Material H**. The results show that DDNM restores both semantic information and the trigger pattern from the transformed images. Our ZIP model can restore semantic information while removing attack patterns**.**
>
> We provide **quantitative** performance comparison as follows. The results reveal that, while DDNM shows better clean accuracy than ZIP, it cannot effectively defend against backdoor attacks like Blended.
>
> |                        | No Defense        | Blur+DDNM             | ZIP (Ours)           |
> | ---------------------- | ----------------- | --------------------- | -------------------- |
> | Imagenette (256 × 256) | CA ↑/ ASR ↓/ PA ↑ | CA ↑/ ASR ↓/ PA ↑     | CA ↑/ ASR ↓/ PA ↑    |
> | BadNet                 | 84.99/94.53/14.98 | **84.56**/9.14/82.24  | 84.05/**7.55/83.97** |
> | Blended                | 86.14/99.85/10.19 | **85.37**/93.37/15.46 | 81.42/**8.35/78.36** |
>
> ***W3:  Evaluation using a different diffusion model, or training on a different dataset?***
>
> To address your concerns, we **use a different diffusion model [1] pre-trained on CIFAR datasets** to defend against BadNet/Blended/PhysicalBA attacks. Our results showcase its effectiveness. Additionally, we observe that the model pre-trained on ImageNet performs better than the model pre-trained on CIFAR, highlighting the importance of pre-training data quality and quantity. In our experiments, we select the ImageNet-pre-trained diffusion model as our backbone, which demonstrates excellent zero-shot purification performance on both in-distribution (Imagenette) and out-of-distribution (CIFAR-10, GTSRB) images, affirming our approach's effectiveness.
>
> |                 | No Defense        | ZIP (pre-trained on CIFAR) | ZIP(pre-trained on ImageNet) |
> | --------------- | ----------------- | --------------------------------------------- | ------------------------------------------------ |
> | CIFAR (32 × 32) | CA ↑/ ASR ↓/ PA ↑ | CA ↑/ ASR ↓/ PA ↑  | CA ↑/ ASR ↓/ PA ↑   |
> | BadNet          | 82.31/99.98/10.00 | 70.35/12.56/66.04   | 78.97/5.53/79.10   |
> | Blended         | 80.26/99.96/10.03 | 70.96/16.39/49.26  | 72.62/7.75/57.98    |
> | PhysicalBA      | 85.3/98.73/11.2   | 80.30/10.05/78.03    | 80.10/4.33/80.33  |
>
> [1] Denoising diffusion probabilistic models. NeurIPS, 2020.
>
> ***W4: More recent baselines should be evaluated***.
>
> Thanks for this comment. We **included contemporaneous work BDMAE [1] as a recent baseline in Section 4.4.1**, published online in March 2023. BDMAE is a black-box defense method that first identifies and masks trigger regions, then restores them using a masked autoencoder.
>
> As demonstrated in Table 3, BDMAE can defend against patch-based attacks like BadNets (ASR=1.12), but it fails in non-patch attacks like Blended (ASR=99.88). In contrast, **our ZIP method can effectively defend against both attacks** (ASR values are low), making it a more practical defense.
>
> It is also worth clarifying that black-box purification in the zero-shot setting is a very challenging task, and as a result, baseline methods are indeed rare.
>
> [1]Mask and Restore: Blind Backdoor Defense at Test Time with Masked Autoencoder. arXiv, 2023.

---

> > ### Comment · Reviewer_xL3D · 2023-08-15
> >
> > Thank the authors for providing responses to address my main concerns. After reading the rebuttal to mine and other reviewers, I would like to raise my score.

---

### Official Review · Reviewer_mm1a · 2023-07-23

**Soundness:** 3 good
**Presentation:** 3 good
**Contribution:** 2 fair
**Rating:** 6
**Confidence:** 5

**Summary:**

This paper proposes a novel backdoor defense framework called "zero-shot image purification" (ZIP) designed to protect against backdoor attacks on real-world black-box models. The proposed ZIP framework consists of two steps: a linear transformation applied to the poisoned image to remove the backdoor pattern and a pre-trained diffusion model used to recover missing semantic information. The reverse process generates high-fidelity purified images without requiring internal information about the poisoned model. The ZIP framework is evaluated on various datasets with different attack types and outperforms state-of-the-art backdoor defense methods. The results are expected to offer valuable insights for future defense methods for black-box models.

**Strengths:**

1. The proposed idea is somewhat novel since the authors leverage the diffusion model to recovery the images. There have been similar works using auto-encoder[1] to reconstruct images to remove backdoor triggers but the exploration of diffusion model is not studied before.

2. The authors provide theoretical analysis of how to modeling image purification with diffusion model.

3. The proposed ZIP shows superior performance than existing defenses.

4. The paper is well-written and easy to follow.

[1]. Y. Liu, Y. Xie, and A. Srivastava, “Neural trojans,” in ICCD, 2017.

**Weaknesses:**

1. The idea of destroying backdoor trigger and reconstructing the images is very similar to pre-processing defenses.  Both defenses aim to remove the backdoor trigger by reconstructing the images.

2. The threat model is not well defined in the paper. The capabilities of defenders and attackers are unclear.

3. The backdoor attacks evaluated in this paper are developed 4-5 years ago, which is a bit too simple and outdated. Many advanced backdoor attacks such as hidden trigger backdoors [2], WaNet [3], LIRA [4], etc. are proposed after that. It would be desirable to evaluate the ZIP against more recent and advanced attacks.

[2]. A. Saha, A. Subramanya, and H. Pirsiavash, “Hidden trigger backdoor attacks,” in AAAI, 2020.
[3]. T. A. Nguyen and A. T. Tran, “Wanet-imperceptible warping-based backdoor attack,” in ICLR, 2021.
[4]. K. Doan, Y. Lao, W. Zhao, and P. Li, “Lira: Learnable, imperceptible and robust backdoor attacks,” in ICCV, 2021.

**Questions:**

Please see weaknesses for details.

**Limitations:**

No potential negative societal impact.

---

> ### Author Rebuttal · Authors · 2023-08-10
>
> ***W1: The idea of destroying backdoor trigger and reconstructing the images is very similar to pre-processing defenses. Both defenses aim to remove the backdoor trigger by reconstructing the images.***
>
> The contribution of our work is different from existing pre-processing defenses [1,2,3,4,5] in the following aspects:
>
> 1. **Zero-shot Capability:** Unlike [1, 2, 3], which requires clean images as training data, our method does not require any prior knowledge of clean/poisoned images.
> 2. **Black-Box Adaptability:** In contrast to [2, 4], which requires model internal information (e.g. gradient, weights), our method only requires model output.
> 3. **Enhanced Semantic Information Preservation:** In comparison to [5], our method can better preserve semantic information as demonstrated in Table 1.
> 4. **Theoretical Justification:** We provide theoretical analyses that justify the efficacy of our RND-based purification approach.
>
> Overall, although sharing similarities with pre-processing defenses, our ZIP framework can work in more challenging and realistic scenarios.
>
> [1]. Y. Liu, Y. Xie, and A. Srivastava, “Neural trojans,” ICCD, 2017.
> [2] H. Qiu, Y. Zeng, S. Guo, T. Zhang, M. Qiu, and B. Thuraisingham, "Deepsweep: An evaluation framework for mitigating DNN backdoor attacks using data augmentation." ASIA CCS, 2021.
> [3] S. Udeshi, S. Peng, G. Woo, L. Loh, L. Rawshan, and S. Chattopadhyay, "Model agnostic defense against backdoor attacks in machine learning."  IEEE Transactions on Reliability, 2022.
> [4] B. G. Doan, E. Abbasnejad, and D. C. Ranasinghe, "Februus: Input purification defense against trojan attacks on deep neural network systems." ACSAC, 2020.
> [5] Y. Li, T. Zhai, Y. Jiang, Z. Li, and S.-T. Xia, "Backdoor attack in the physical world." ICLR workshop, 2021.
>
> ***W2: The threat model is not well defined in the paper. The capabilities of defenders and attackers are unclear.***
>
> Thanks for raising this question. We focus on backdoor defense in the black-box setting, where the **defender** has only access to the poisoned model output, without access to the model’s internal parameters or training datasets. On the other hand, the **attackers** can access and modify model internal components, training datasets, or any other necessary information to implement their attacks. We will add a paragraph introducing the threat model in the next version.
>
> ***W3: The backdoor attacks evaluated in this paper are developed 4-5 years ago, which is a bit too simple and outdated. Many advanced backdoor attacks such as hidden trigger backdoors [2], WaNet [3], LIRA [4], etc. are proposed after that. It would be desirable to evaluate the ZIP against more recent and advanced attacks.***
>
> For more advanced attacks, we conduct experiments with the **WaNet [1], Blind [2], and Label-Consistence attack [3]** on Imagenette datasets to further demonstrate our defense effectiveness. The quantitative results are listed below, while the qualitative results are provided in the rebuttal pdf. The results show that our ZIP can successfully defend against advanced attacks.
>
> |                        | No Defense         | ShrinkPad (defense) | Blur (defense)    | ZIP (Ours)           |
> | ---------------------- | ------------------ | ------------------- | ----------------- | -------------------- |
> | Imagenette (256 × 256) | CA ↑/ ASR ↓/ PA ↑  | CA ↑/ ASR ↓/ PA ↑   | CA ↑/ ASR ↓/ PA ↑ | CA ↑/ ASR ↓/ PA ↑    |
> | WaNet                  | 85.45 /97.93/11.84 | 73.63 /26.87/60.35  | 62.98/17.88/56.81 | **79.13/7.77/76.78** |
> | Blind                  | 79.15/99.64/10.42  | 65.22/6.34/65.60    | 75.26/28.12/67.54 | **78.80/5.32/79.03** |
> | LabelConsistent        | 77.57/73.17/36.56  | 66.14/0.25/64.53    | 50.36/0.28/50.14  | **73.63/0.12/74.26** |
>
> [1] Nguyen, Anh, and Anh Tran. "Wanet--imperceptible warping-based backdoor attack."  ICLR, 2021.
> [2] Bagdasaryan, Eugene, and Vitaly Shmatikov. "Blind backdoors in deep learning models." USENIX Security, 2021.
> [3] Turner, Alexander, Dimitris Tsipras, and Aleksander Madry. "Label-consistent backdoor attacks." arXiv 2019.

---

> > ### Comment · Reviewer_mm1a · 2023-08-15
> >
> > I've read the authors rebuttal. Most of my concerns are adequately addressed with clarification and additional experimental results. Thus, I'd like to change my rating to weak accept.

---

### Official Review · Reviewer_L743 · 2023-07-25

**Soundness:** 2 fair
**Presentation:** 3 good
**Contribution:** 2 fair
**Rating:** 5
**Confidence:** 4

**Summary:**

The work deals with the challenge of defending against backdoor attacks for black-box models in zero-shot settings using pre-trained diffusion models. Based on some theoretical insights, the proposed method modifies the reverse process of the diffusion model so the recovered images can be more high-fidelity and clean. Experimental results show that the proposed framework is effective in defending against backdoor attacks without prior knowledge of the model or additional retraining.

**Strengths:**

- The model-agnostic nature of the proposed method enhances its practicality as it can be adopted by any model without retraining.

- The theoretical analysis is clear and comprehensive.

- The additional techniques for speeding up the algorithm make the framework more realistic and applicable.

**Weaknesses:**

- Previous works have shown that Diffusion models can be used for image purification [1, 2, 3]. Thus, the paper’s novelty primarily lies in the adaptation of the traditional diffusion model for high-fidelity image recovery. While these improvements are theoretically driven, empirical evidence demonstrating the effectiveness of these enhancements or validating the proposed theories is lacking.

- The experiments adopt three backdoor attacks: BadNet, PhysicalBA, and Blended. But more advanced attacks [4, 5] have been proposed and adopted in recent backdoor defense literature [6].


- There is an inconsistency in the statement on L344-345: “However, by switching to a different transformation, such as solely Blur or Grayscale, the attacks can be effectively mitigated.” ([pdf](zotero://open-pdf/library/items/HJZXQTNN?page=9)) In contrast, Table 4 reveals subpar defense performance when solely employing Grayscale.


[1] Nie et al. Diffusion Models for Adversarial Purification.

[2] Wang et al. Guided Diffusion Model for Adversarial Purification.

[3] Sun et al. PointDP: Diffusion-driven Purification against Adversarial Attacks on 3D Point Cloud Recognition.

[4] Bagdasaryan and Shmatikov. Blind Backdoors in Deep Learning Models.

[5] Nguyen and Tran. WaNet -- Imperceptible Warping-based Backdoor Attack.

[6] Doan et al. Defending Backdoor Attacks on Vision Transformer via Patch Processing.

**Questions:**

- Regarding the setting of 4.4.2, if the attacker assumes that the defender is using Blur for transformation, will switching to a different transformation, such as Grayscale, be effective?

- The statement "even when using an attacked model as the classifier" on L66 is unclear. Could you provide more context or clarification for this scenario?

**Limitations:**

The authors have discussed the limitations of their work. Please see the Weaknesses stated above to see other limitations I find.

---

> ### Author Rebuttal · Authors · 2023-08-10
>
> ***W1: Empirical evidence supporting the proposed enhancements to [1,2,3] and validating theories is currently lacking.***
>
> Thanks for the thoughtful question. We would like to clarify the novelty and contribution of our work as follows.
>
> 1. **An effective black-box defense against backdoor attacks:**  Existing methods [1, 2, 3] rely on the denoising capability of diffusion models (e.g., DDPM), which is limited against backdoor patterns (e.g., patch-based patterns) as demonstrated in [4]. In contrast, our method focuses on defending against backdoor attacks, especially under **the novel and challenging black-box scenario**.
> 2. **A novel reverse diffusion process**: Previous methods [1, 3] employ naive diffusion models that could produce unpredictable purified images, while we innovatively integrate RND theory into the reverse diffusion process, ensuring high-fidelity recovery of images and eliminating backdoor patterns. To empirically demonstrate the advantages, we implement DiffPure [1] and its combination with Blur transformation (Blur+DiffPure) as baselines. As shown in the table below, empirical results show ZIP performs better than DiffPure and Blur+DiffPure.
>
> 3. **Linear transformation for backdoor destruction:** Linear transformation is a basic step in the RND theory. We explore the effectiveness of linear transformations, such as Blur and Grayscale, in destroying backdoor patterns. Empirical results of linear transformation effectiveness are provided in Sec 4.4.1.
>
> |                        | No Defense        | DiffPure          | Blur+DiffPure     | ZIP (Ours)           |
> | ---------------------- | ----------------- | ----------------- | ----------------- | -------------------- |
> | Imagenette (256 × 256) | CA ↑/ ASR ↓/ PA ↑ | CA ↑/ ASR ↓/ PA ↑ | CA ↑/ ASR ↓/ PA ↑ | CA ↑/ ASR ↓/ PA ↑    |
> | BadNet                 | 84.99/94.53/14.98 | 78.85/91.41/18.11 | 75.13/10.16/74.31 | **84.05/7.55/83.97** |
> | Blended                | 86.14/99.85/10.19 | 80.63/43.82/56.68 | 75.89/13.53/73.98 | **81.42/8.35/78.36** |
>
> [1] Diffusion models for adversarial purification. ICML, 2022.
> [2]  Guided diffusion model for adversarial purification. arXiv, 2022.
> [3] Pointdp: Diffusion-driven purification against adversarial attacks on 3d point cloud recognition.  arXiv, 2022.
> [4] DIFFender: Diffusion-Based Adversarial Defense against Patch Attacks in the Physical World.  arXiv, 2023.
>
> ***W2: Adopt more recent attacks like Blind attack and WaNet attack .***
>
> For more recent attacks, we have conducted experiments with Blind [1], WaNet [2], and Label-Consistence [3] on Imagenette datasets to further demonstrate our defense effectiveness. The quantitative results are listed below, and the qualitative results are provided in the rebuttal PDF. The results show that our ZIP can successfully defend against various advanced attacks.
>
> |                        | No Defense        | ShrinkPad (defense) | Blur (defense)    | ZIP (Ours)           |
> | ---------------------- | ----------------- | ------------------- | ----------------- | -------------------- |
> | Imagenette (256 × 256) | CA ↑/ ASR ↓/ PA ↑ | CA ↑/ ASR ↓/ PA ↑   | CA ↑/ ASR ↓/ PA ↑ | CA ↑/ ASR ↓/ PA ↑    |
> | WaNet                  | 85.45/97.93/11.84 | 73.63/26.87/60.35   | 62.98/17.88/56.81 | **79.13/7.77/76.78** |
> | Blind                  | 79.15/99.64/10.42 | 65.22/6.34/65.60    | 75.26/28.12/67.54 | **78.80/5.32/79.03** |
> | LabelConsistent        | 77.57/73.17/36.56 | 66.14/0.25/64.53    | 50.36/0.28/50.14  | **73.63/0.12/74.26** |
>
> [1] Blind backdoors in deep learning models. USENIX Security, 2021.
> [2] Wanet--imperceptible warping-based backdoor attack.  ICLR, 2021.
> [3] Label-consistent backdoor attacks. arXiv 2019.
>
> ***W3: Inconsistency noted in L344-345***
>
> In Table 4, the effectiveness of Grayscale is shown in its acceptable performance against enhanced Blended attacks. However, we agree that GrayScale shows subpar performance in some cases. To avoid further confusion, we will revise the expression in the next version. Thank you for pointing this out.
>
> Moreover, we would like to emphasize that Table 4 is the result of **enhanced attacks**, where the purified poison images are selected as new attack images to inject backdoors. Existing backdoor defense methods [1] fail to defend against such enhanced attacks. In comparison, our proposed ZIP can defend against the enhanced attack if proper transformation is applied.
>
> [1] Rethinking the Trigger of Backdoor Attack. arXiv, 2021
>
> ***Q1: Regarding the setting of 4.4.2, if the attacker assumes that the defender is using Blur for transformation, will switching to a different transformation, such as Grayscale, be effective?***
>
> Yes, the defense is expected to be effective when the attacker uses grayscale to enhance their attack while the defender uses Blur. As shown in Table 4, the defense is effective when the attacker enhances their attack with different transformations as the defender. In practical scenarios, the defender's transformation can vary and be hidden from potential attackers. This characteristic enhances the robustness of our defense.
>
> ***Q2: The statement "even when using an attacked model as the classifier" on L66 is unclear. Could you provide more context or clarification for this scenario?***
>
> The “attacked model” is a classifier already poisoned by backdoor attacks (e.g., BadNets). We use the attacked model as the downstream classifier following the typical settings in previous work on purification-based defense [1, 2, 3]. We will revise the expression in the next version to avoid confusion.
>
> [1] Backdoor attack in the physical world. ICLR workshop, 2021.
> [2] Februus: Input Purification Defense Against Trojan Attacks on Deep Neural Network Systems.  ACSAC, 2020.
> [3] Deepsweep: An evaluation framework for mitigating DNN backdoor attacks using data augmentation. ASIA CCS, 2021.

---

> > ### Comment · Reviewer_L743 · 2023-08-13
> > **Follow-up questions**
> >
> > Thank you for your effort to respond to my comments.
> >
> > > To empirically demonstrate the advantages, we implement DiffPure [1] and its combination with Blur transformation (Blur+DiffPure) as baselines.
> >
> > Can you please describe in detail the difference between Blur+DiffPure and your ZIP method in practice?
> >
> > > Yes, the defense is expected to be effective when the attacker uses grayscale to enhance their attack while the defender uses Blur.
> >
> > Can you show the quantitative result of the experiment?

---

> > > ### Author Response · Authors · 2023-08-15
> > > **Response to the follow-up questions**
> > >
> > > **Q1: Can you please describe in detail the difference between Blur+DiffPure and your ZIP method in practice?**
> > >
> > > Our ZIP can recover the semantic information removed by the Blur, while Blur+DiffPure can not. In detailed comparison:
> > >
> > > 1. **Blur+DiffPure:** DiffPure aims to remove attack patterns by first diffusing images with noise and then recovering images through an **unconditional reverse process**. While coupling DiffPure with Blur enhances its trigger removal capability (lower ASR), its unconditional generative process fails to effectively recover the semantic information removed by Blur. This leads to a drop in clean accuracy (CA). In the above table, Blur+DiffPure exhibits poorer CA compared to both DiffPure and our ZIP.
> > >
> > > 2. **ZIP:** In contrast, our ZIP, after Blur, utilizes a **conditional reverse process** to recover semantic information deleted by Blur through RND theory. Specifically, using RND, the purified image $\hat{\mathbf{x_t}}$ at time $t$ is approximated as follows:
> > > $$\hat{\mathbf{x_t}} = \sqrt{\bar{\alpha_t}}\mathbf{A}^{\dagger}\mathbf{x^A} +(\mathbf{I}-\mathbf{A}^{\dagger} \mathbf{A}) \mathbf{x_t} + \mathbf{A}^{\dagger}\mathbf{A}\sqrt{1-\bar{\alpha_t}}\boldsymbol{\epsilon_t},$$
> > >
> > >    where the blurred image $\mathbf{x}^A$ and diffusion output $\mathbf{x}_t, \boldsymbol{\epsilon}_t $ collaboratively generate high-fidelity purified images. We proved in Corollary 3.2.1 that our proposed ZIP preserves the semantic information, and our experiments also proved that empirically (Best CA with ZIP).
> > >
> > > **Q2: Can you show the quantitative result of the experiment?**
> > >
> > > Yes, the experiments where the attacker uses grayscale to enhance their attack while the defender uses Blur as defense transformation are below.
> > >
> > > We can observe that when the defender uses the same transformation (Grayscale) as the attack, the defense against enhanced attack tends to fail. However, switching to a different transformation like Blur enables the defense to succeed.
> > >
> > > | Enhanced Attack(using Grayscale during attack)        | Original Trigger  | ZIP (Blur+Grayscale) | ZIP (Blur)        | ZIP (Grayscale)   |
> > > | ---------------------- | ----------------- | -------------------- | ----------------- | ----------------- |
> > > | Imagenette (256 × 256) | CA ↑/ ASR ↓/ PA ↑ | CA ↑/ ASR ↓/ PA ↑    | CA ↑/ ASR ↓/ PA ↑ | CA ↑/ ASR ↓/ PA ↑ |
> > > | BadNets                | 86.64/87.36/21.98 | 80.38/19.28/72.45    | **82.44/7.76/83.38**  | 70.77/69.36/34.21 |
> > > | Blended                | 84.89/99.59/10.44 | **81.93**/39.81/59.43    | 81.50/**11.29/77.12** | 77.01/98.98/10.82 |
> > >
> > >
> > > Thanks again for your effort in helping us improve our work. And we kindly ask for your reconsideration of the score in light of our clarifications.

---

> > > > ### Author Response · Authors · 2023-08-18
> > > >
> > > > Dear Reviewer L743,
> > > >
> > > > We hope our response has addressed your concerns. If you need further clarification or have additional questions, please let us know. We value your input in refining our work.
> > > >
> > > > Best,

---

### Author Rebuttal · Authors · 2023-08-10

# Global Response to All Reviewers

Thank you for your time and efforts in reviewing our work. We greatly appreciate reviewers’ recognition of the quality and novelty of our research. Here is a summary of our response.

**Related Work:**

Our paper addresses the challenge of achieving practical **zero-shot purification defense in a black-box setting.** Our ZIP method introduces novelty compared to existing approaches in the following ways:

1. Compared to **pre-processing defenses**, our ZIP does not require access to poisoned model internals or prior knowledge of clean/poisoned images, while still maintaining better semantic information after purification.
2. Compared to **diffusion-based purification models**, ZIP enhances trigger destruction with linear transformation and incorporates RND theory to improve our reverse diffusion process.
3. Compared to **black-box detection methods**, our ZIP can remove backdoor effects from poisoned images through purification, and provides a theoretical analysis of its effectiveness.

In summary, our model presents a novel and robust defense against various attacks in more complex scenarios.

**Experiments:**

We have added the following experiments:

1. Defense Performance against **WaNet, Blind, and Lable-Consistent** attacks: In Rebuttal_Table 1, ZIP shows effective defense performance with three attacks, which showcases that our ZIP can defend against recent advanced attacks.
2. Performance comparisons with **image purification method**: DiffPure, and Blur+DiffPure: In Rebuttal_Table 2, ZIP shows superior defense performance compared to DiffPure and Blur+DiffPure, which validates the proposed enhancements.
3. Defense Performance with **different pre-trained diffusion models**: In Rebuttal_Table 3, ZIP shows effective defense using a diffusion model pre-trained on CIFAR, demonstrating that ZIP can work with different diffusion models.
4. Defense Performance with **classifiers in different architectures**: In Rebuttal_Table 2 and 4, ZIP shows its effective defense performance with poisoned classifiers in ResNet-34 and VGG-16 networks.
5. Performance comparisons with **image restoration method**: DDNM: In Rebuttal_Table 2, ZIP shows better defense performance compared to DDNM, validating our superiority.

We also would like to mention that:

6. We include **contemporaneous work BDMAE** as a recent defense baseline in our Section 4.4.1, and experiments show that our ZIP method can effectively defend against BadNet and Blended attack, while BDMAE cannot.
7. Qualitative performance comparisons with **DDNM** are provided in Supplementary Material H, where ZIP shows better performance in removing trigger patterns.
8. Extensive **visualization results of ZIP** purification are provided in Supplementary B to validate our proposed design.

In summary, the experiments show that **our ZIP can defend against advanced backdoors in diverse forms,** and ZIP shows better defense performance **compared to various baselines** including BDMAE, DiffPure, and DDNM.

We really appreciate your constructive comments, as they help us significantly improve the quality of our paper. We sincerely hope our response can address your concerns.

---

### Decision · Program_Chairs · 2023-09-21

**Decision:**

Accept (poster)

**Comment:**

According to the comments from the reviewers who has engaged in discussions during the rebuttal period, some of the reviewers have raised the scores and most comments have been well-addressed. The main strength of this paper is its sufficient novelty and comprehensive and clear theoretical analysis. Based on the above reasons, the AC tends to recommend the acceptance of this paper.